# Structural basis of α$_{1A}$-adrenergic receptor activation and recognition by an extracellular nanobody

Yosuke Toyoda [1,2,7,8] ✉, Angqi Zhu [2,3,8], Fang Kong [2,3], Sisi Shan[1,2,4], Jiawei Zhao[1,2], Nan Wang[2,3], Xiaoou Sun[1,2], Linqi Zhang[1,2,4], Chuangye Yan [2,3] ✉, Brian K. Kobilka [5] ✉ & Xiangyu Liu [2,6] ✉

The α$_{1A}$-adrenergic receptor (α$_{1A}$AR) belongs to the family of G protein-coupled receptors that respond to adrenaline and noradrenaline. α$_{1A}$AR is involved in smooth muscle contraction and cognitive function. Here, we present three cryo-electron microscopy structures of human α$_{1A}$AR bound to the endogenous agonist noradrenaline, its selective agonist oxymetazoline, and the antagonist tamsulosin, with resolutions range from 2.9 Å to 3.5 Å. Our active and inactive α$_{1A}$AR structures reveal the activation mechanism and distinct ligand binding modes for noradrenaline compared with other adrenergic receptor subtypes. In addition, we identified a nanobody that preferentially binds to the extracellular vestibule of α$_{1A}$AR when bound to the selective agonist oxymetazoline. These results should facilitate the design of more selective therapeutic drugs targeting both orthosteric and allosteric sites in this receptor family.

Adrenergic receptors (ARs), which mediate physiological responses to the neurotransmitter noradrenaline (norepinephrine) and the hormone adrenaline (epinephrine), are family A G protein-coupled receptors (GPCRs). They mediate responses to sympathetic nervous system activation and are subdivided into α$_1$ (α$_{1A}$, α$_{1B}$ and α$_{1D}$), α$_2$ (α$_{2A}$, α$_{2B}$ and α$_{2c}$) and β (β$_1$, β$_2$, and β$_3$) ARs. α$_1$ARs are predominantly coupled to the heterotrimeric G$_{q/11}$ family of G proteins, leading to the activation of phospholipase C and the increase of cytosolic Ca$^{2+}$. α$_{1A}$AR was cloned from the bovine brain and initially designated α$_{1C}$AR[1]. As the original α$_{1A}$AR and α$_{1D}$AR appeared to represent the same subtypes, these clones have been renamed α$_{1A}$AR (formerly α$_{1C}$AR), α$_{1B}$AR (formerly α$_{1B}$AR) and α$_{1D}$AR (formerly α$_{1A}$AR and α$_{1D}$AR)[2,3]. α$_1$ARs are expressed in a wide range of tissues including blood vessels, kidney, spleen, liver, brain and lower urinary tract[2,3]. In the periphery, postsynaptic α$_1$AR activation mediates smooth muscle contraction, therefore the selective α$_{1A}$AR agonist oxymetazoline[4] is clinically used for the treatment of nasal congestion, whereas selective α$_{1A}$AR antagonists such as tamsulosin and silodosin are prescribed to treat hypertension and benign prostatic hyperplasia[5] (Supplementary Fig. 1). As α$_1$AR plays a role in regulating synaptic plasticity and memory consolidation[2], the α$_1$AR antagonist prazosin is used to reduce nightmares and overall Post Traumatic Stress Disorder symptoms[6] and has a potential for the preventing cytokine storm syndrome caused by the severe acute respiratory syndrome coronavirus 2, a leading cause of morbidity and mortality in coronavirus disease 2019[7].

[1]School of Medicine, Tsinghua University, Beijing 100084, China. [2]Beijing Frontier Research Center for Biological Structure, Beijing Advanced Innovation Center for Structural Biology, Tsinghua University, Beijing 100084, China. [3]State Key Laboratory of Membrane Biology, Tsinghua-Peking Center for Life Sciences, School of Life Sciences, Tsinghua University, Beijing 100084, China. [4]NexVac Research Center, Comprehensive AIDS Research Center, Center for Infectious Disease Research, Tsinghua University, Beijing 100084, China. [5]Department of Molecular and Cellular Physiology, Stanford University School of Medicine, Stanford, CA 94305, USA. [6]State Key Laboratory of Membrane Biology, Tsinghua-Peking Center for Life Sciences, School of Pharmaceutical Sciences, Tsinghua University, Beijing 100084, China. [7]Present address: Institute for Integrated Cell-Material Sciences, Institute for Advanced Study, Kyoto University, Kyoto 606-8501, Japan. [8]These authors contributed equally: Yosuke Toyoda, Angqi Zhu. ✉e-mail: toyoda.yosuke.2r@kyoto-u.ac.jp; yancy2019@mail.tsinghua.edu.cn; kobilka@stanford.edu; liu_xy@mail.tsinghua.edu.cn

Recent progress in the structural characterization of GPCRs including the adrenergic receptors has clarified the mechanism of ligand recognition and G protein activation. The βARs are extensively well-characterized GPCRs and a number of structures have been determined in the active and inactive states[8–13]. Moreover, recent structures of active and inactive α2AAR[14,15], active α2BAR-Gi/o[16] and inactive α2CAR[17] demonstrated the subtype selectivity of ligand recognition between α2ARs and βARs. However, little is known about the structure and mechanism of activation of the α1AR subtypes. The only available structure is the inactive α1BAR structure[18]. Here we present three cryo-electron microscopy (cryo-EM) structures of active α1AAR bound to oxymetazoline and the endogenous agonist noradrenaline, along with inactive α1AAR bound to tamsulosin. We also discovered a nanobody (a single domain antibody) Nb29 that binds to the extracellular vestibule of the agonist-binding pocket. Antibodies against GPCRs have attracted particular interest for pharmaceutical applications[19], and are useful research tools for stabilizing GPCR conformations for structural analysis[20,21]. Nevertheless, only a few class A GPCR structures in complex with the extracellular antibody are available[22–26]. These findings may guide the development of more effective drugs for the α1AAR.

## Results

### Structure determination of active and inactive α1AAR

We expressed human α1AAR in baculovirus-infected *Spodoptera frugiperda* (*Sf*9) insect cells. We constructed a variant of human α1AAR lacking residues 371–466 in the C-terminus, and three *N*-linked glycosylation sites (N7Q, N13Q and N22Q) in the N-terminus were mutated. To further stabilize the receptor, we discovered a conformationally selective nanobody from a library of synthetic nanobodies displayed on the surface of *Saccharomyces cerevisiae*[27]. After two rounds of magnetic-activated cell sorting (MACS) and four rounds of fluorescence-activated cell sorting (FACS) with oxymetazoline- and

tamsulosin-bound α1AAR, we identified Nb29 as the most enriched clone (Fig. 1a and Supplementary Fig. 2; See Method). An on-yeast titration assay indicated that Nb29 has selectivity for oxymetazoline-bound α1AAR compared with apo, noradrenaline-, and antagonist (tamsulosin and phentolamine)-bound states (Fig. 1b and Supplementary Table 1a). In a ligand binding assay, Nb29 induced the left shift of the agonist competition curves for α1AAR over α1B- and α1DAR (Fig. 1c and Supplementary Table 1b). Although Nb29 on its own competed for the antagonist [³H]prazosin binding for α1AAR and might affect the competition binding results, we observed a larger left shift of oxymetazoline competition curves than those for noradrenaline, which is in agreement with the on-yeast titration result (Fig. 1d, e and Supplementary Table 1c).

To solve the structure, we formed the α1AAR complex with Nb29 and obtained the initial cryo-EM structure at 4 Å resolution, and found that Nb29 binds to the extracellular region of the α1AAR. To further stabilize the receptor, we used α1AAR construct fused with a minimal-T4 lysozyme (mT4L) in the intracellular loop 3 (ICL3)[28] (Supplementary Fig. 3a), and formed a complex with engineered minimal Gsq protein (miniGsq) in which mini-Gs was substituted with the 15 residues of carboxyl-terminal α5 helix of Gq protein[29] (Methods and Supplementary Fig. 3b, c). The α1AAR in complex with the heterotrimeric Gq/11 proteins was not stable enough for structure determination. Finally, we obtained the cryo-EM structures of active α1AAR bound with oxymetazoline (Nb29-α1AAR-miniGsq) at a global resolution of 2.9 Å (Fig. 2a, b; Supplementary Figs. 3d–f and 4a; Supplementary Table 2). The cryo-EM map allowed the model building of most of the regions with a clear electron density for the ligand. The map density of the extracellular region of α1AAR is relatively clear because of the bound Nb29, whereas there is poor density for the fused mT4L due to map refinement by masking out the mT4L. Subsequently, we also solved the Nb29-α1AAR-miniGsq complex bound with noradrenaline at a global resolution of 3.5 Å (Fig. 2c, d;

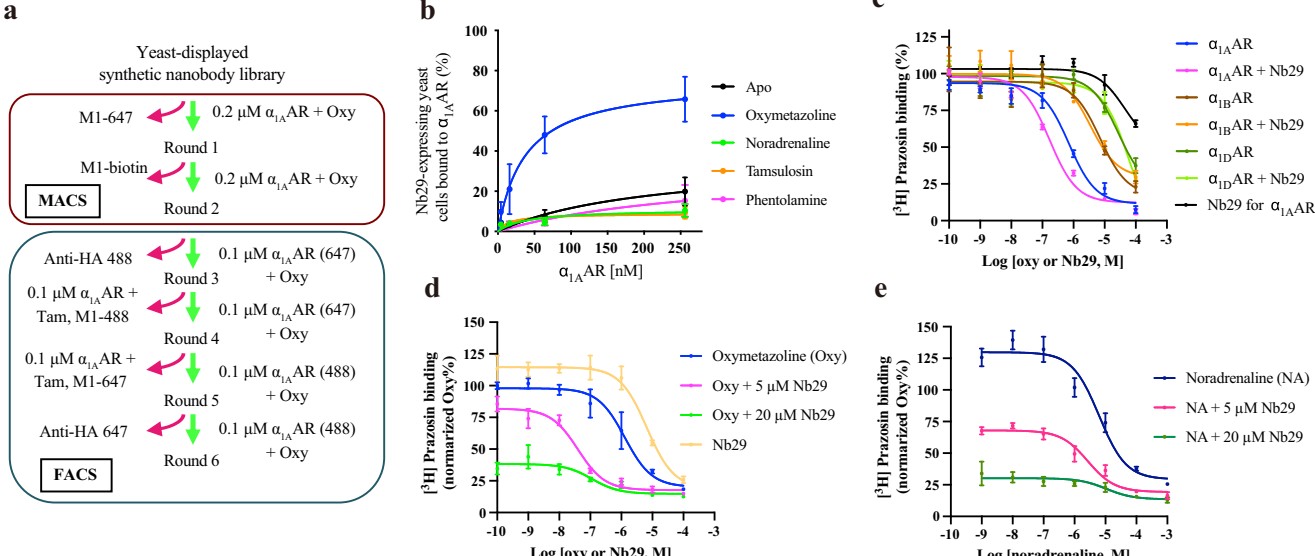

**Fig. 1 | Selection and characterization of Nb29 as a conformationally selective nanobody for α1AAR. a** Flow chart of the selection process of conformationally selective nanobodies from the yeast-displayed nanobody library. For rounds 1 and 2, 0.2 μM α1AAR bound to oxymetazoline (oxy) was used for selection and alexa647-labeled anti-FLAG M1 antibody (M1-647) or biotin-labeled anti-FLAG M1 antibody Fab fragment (M1-biotin) was used for the preclear. For the FACS selection, different combinations of counterselection were performed using oxymetazoline, tamsulosin (Tam), M1-488/647, and anti-HA antibodies. **b** On-yeast titration to estimate the affinity of Nb29 for α1AAR, evaluated by flow cytometry. The ratio of Nb29-displayed yeast cells bound purified α1AAR in the presence or absence of

500 μM ligands was analyzed. The data represent mean ± s.e.m. of *n* = 3 independent measurements. **c** ³H-prazosin radioligand competition binding of α1AR subtype for oxymetazoline in *Sf*9 membranes. Samples in the presence of Nb29 were used at 5 μM concentration of Nb29. The data represent mean ± s.e.m. of *n* = 3 independent measurements. **d, e** ³H-prazosin radioligand competition binding of the purified α1AAR-bound M1-Flag affinity resin for oxymetazoline (Oxy), noradrenaline (NA) or Nb29. The data represent mean ± s.e.m. of *n* = 3 (NA + 5 μM Nb29), and *n* = 6 (the others) independent measurements. Binding affinity values are provided in Supplementary Table 1. Source data are provided in the Source Data file.

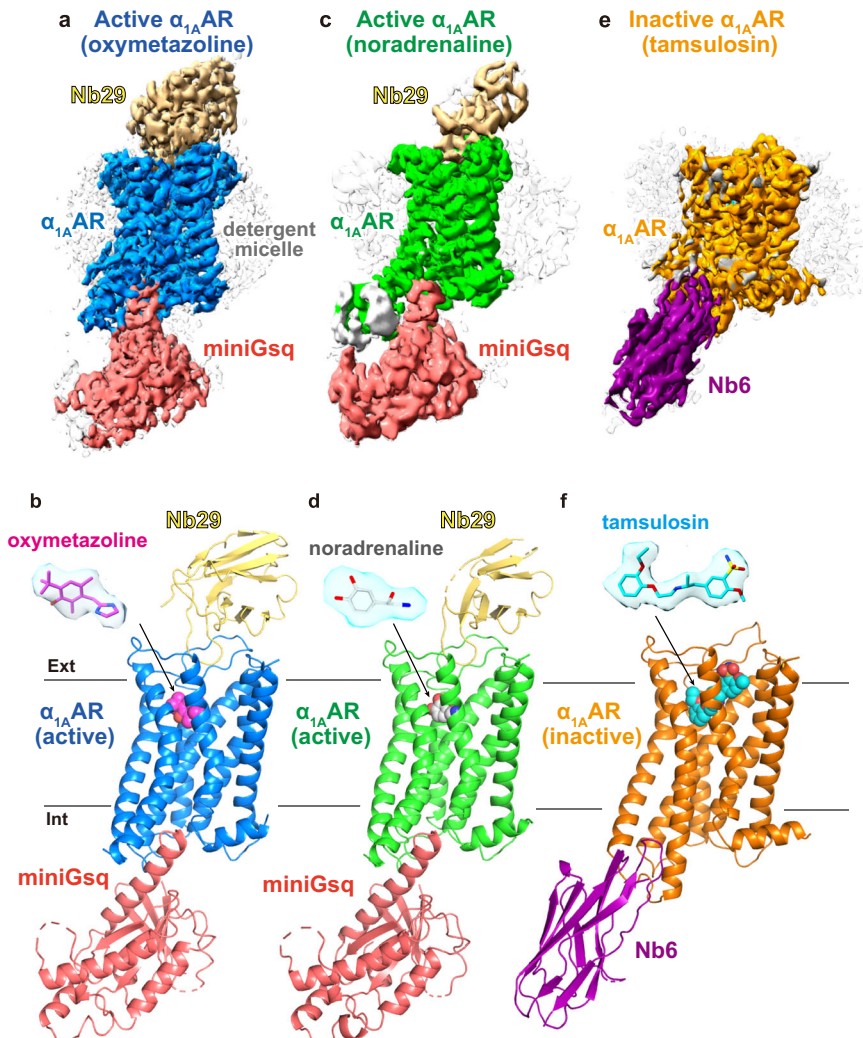

**Fig. 2 | Cryo-EM structures of oxymetazoline- and noradrenaline-bound Nb29–α$_{1A}$AR-miniGsq and tamsulosin-bound α$_{1A}$AR-Nb6 complexes.** The cryo-EM density maps and structure models of the Nb29-α$_{1A}$AR-miniGsq complexes bound to the agonists oxymetazoline (**a, b**) and noradrenaline (**c, d**), and α$_{1A}$AR-Nb6 complex bound to the antagonist tamsulosin (**e, f**). The detergent micelle (**a, c, e**) and unmodelled mT4L (**b**) are shown in gray. The densities of the ligands (shown as sticks) are depicted as surfaces. Color code for the proteins is as follows: oxymetazoline-bound active α$_{1A}$AR (blue), noradrenaline-bound active α$_{1A}$AR (green), inactive α$_{1A}$AR (orange), miniGsq (pink), Nb29 (yellow), and Nb6 (purple). Small molecules are colored as follows: oxymetazoline in magenta, noradrenaline in gray, and tamsulosin in cyan.

Supplementary Figs. 3g–i and 4b; Supplementary Table 2). Although the map quality allowed the model building of the receptor with clear electron density for the ligand, the map density for Nb29 is weak. The weaker map density is in agreement with the observation that Nb29 preferentially binds to oxymetazoline-bound receptors over noradrenaline-bound receptors (Fig. 1b). The resolution of the Nb29-dissociated α$_{1A}$AR-miniGsq complex is much lower (~6 Å) than that of the Nb29-bound complex (Supplementary Fig. 3h).

To solve the inactive α$_{1A}$AR structure, our crystallographic and cryo-EM experiments using fusion protein and/or the other nanobodies selected from the synthetic nanobody library were unsuccessful. Thus, we utilized a recently described engineering strategy to enable the binding of nanobody 6 (Nb6) that engages the ICL3 of the inactive-state κ-opioid receptor (kOR)[21]. Based on the cryo-EM structure of engineered neurotensin receptor 1 (NTSR1)-Nb6 complex, we swapped the same site from C205[5.59] of transmembrane (TM) 5 (T247[5.59] of kOR) to G275[6.38] of TM6 (L277[6.38] of kOR) including intracellular loop (ICL) 3 (superscripts indicate Ballesteros-Weinstein numbering for GPCRs[30]). In addition, we introduced two thermo-stabilizing point mutations, S113R[3.39] to mimic allosteric sodium ion

binding[31], and M115W[3.41] to increase expression[32]. These mutations were used for solving other inactive GPCR structures such as prostaglandin E receptor EP4 for the R[3.39] mutation[22], and dopamine D2 receptor for double mutation of positions 3.39 and 3.41[25]. In radio-oligand binding studies, the α$_{1A}$AR-kOR mutant showed enhancement of [³H]prazosin binding by Nb6, but did not significantly alter the binding affinities for the agonist oxymetazoline or the antagonist tamsulosin (Supplementary Fig. 5a, b). We formed the α$_{1A}$AR-Nb6 complex and solved the cryo-EM structure of the inactive α$_{1A}$AR bound to tamsulosin at a global resolution of 3.3 Å resolution. (Fig. 2e–f; Supplementary Fig. 5c–h; Supplementary Table 2). The cryo-EM map allowed the model building of most of the regions with a clear electron density for tamsulosin; the map density of the inactive α$_{1A}$AR-Nb6 complex is relatively weak for ECL2 and well-defined in the cytoplasmic region including Nb6 binding domain where ICL3 and the cytoplasmic sides of TM5 and 6 were exchanged to those of the kOR. Although Nb6 binds to a similar site as in the kOR-Nb6 complex[33] and locks the conformation of the TM5 and TM6 of α$_{1A}$AR, we observed fewer polar interactions in our structure when compared with the kOR-Nb6 and NTSR1-Nb6 complexes[21]

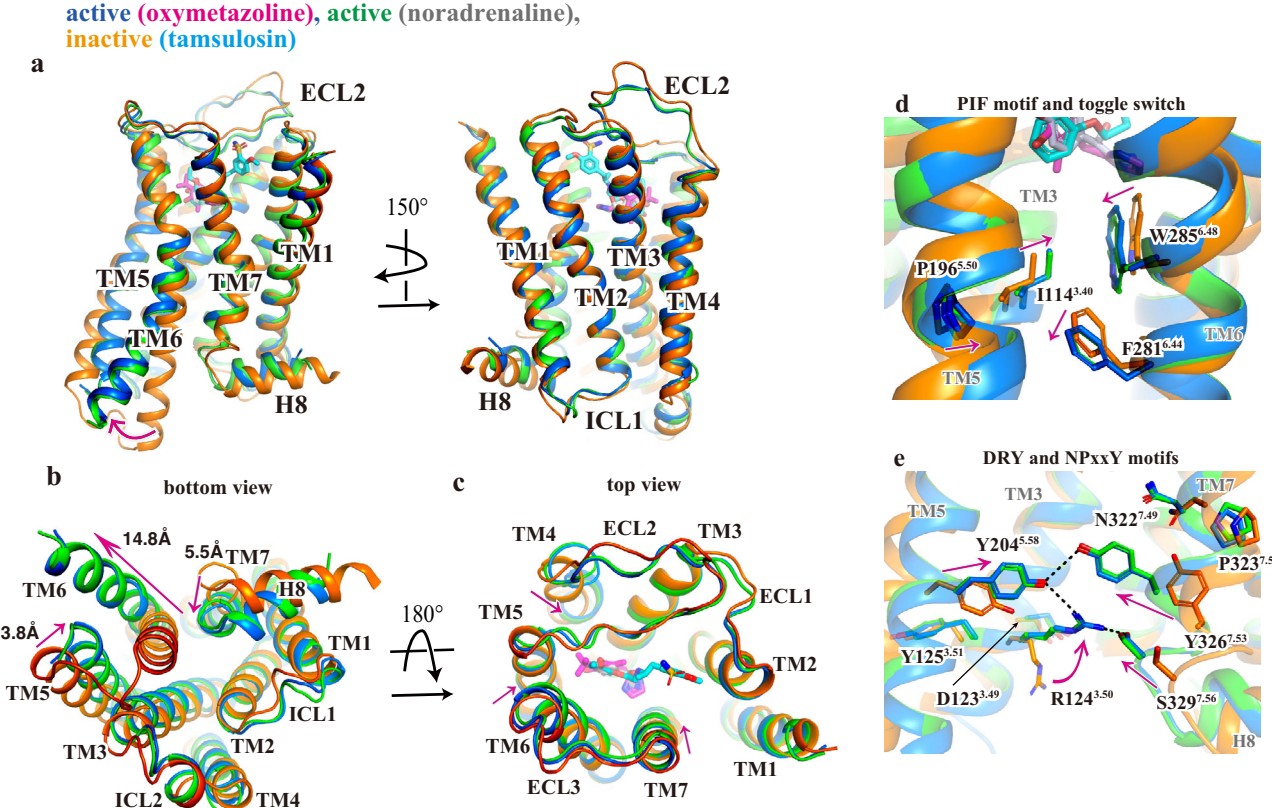

**Fig. 3 | Structural comparison between active and inactive $\alpha_{1A}AR$. a** Comparison of $\alpha_{1A}AR$ between oxymetazoline-bound (blue), noradrenaline-bound (green), and inactive (orange) states viewed from the side. **b** Intracellular view of superposed $\alpha_{1A}ARs$. Distances were measured between the C$\alpha$ atoms of E269[6.30] (E269L[6.30] in $\alpha_{1A}AR$-kOR) in TM6, K212[5.66] in TM5, and C328[7.55] in TM7. **c** Extracellular view of superposed $\alpha_{1A}ARs$. The maximum distances of the side chain displacement of W285 (position 7th carbon of the indole ring) are 2.4 Å between the noradrenaline-bound active state and the tamsulosin-bound inactive state, and 1.8 Å between the oxymetazoline-bound state and the tamsulosin-bound state. **d** Conformational change of PIF motif and toggle switch W285[6.48]. **e** Conformational change of DRY and NpxxY motifs. Conformational changes upon activation are shown with magenta arrows. Hydrogen bonds are shown as black dashed lines.

(Supplementary Fig. 6a–e). The active $\alpha_{1A}AR$ structures enable us to model a putative cholesteryl hemisuccinate (CHS) molecule bordering TM3-5 in the active structures, in contrast, we observed only a weak density in the inactive $\alpha_{1A}AR$ structure, since the side chain of M115W[3.41] overlaps the regions corresponding to the lipid tail of CHS (Supplementary Fig. 6g, h). The side chain of S113R[3.39] is located in the putative sodium ion binding site as designed (Supplementary Fig. 6i, j)[22,31].

Structural comparison of active and inactive states of $\alpha_{1A}AR$ exhibits 14.5 Å outward displacement of an intracellular segment of TM6 that is characteristic of receptor activation (Fig. 3a–c). The TM6 movement is accompanied by a small rotation of the helix, as well as inward movements of TMs 3, 5, and 7 toward TM6. $\alpha_{1A}AR$ also exhibits other characteristics of the activation of class A GPCRs[9,14,16]. We observed the displacement of the side chain of W285[6.48] (Fig. 3d), a highly conserved residue that contributes to conformational changes associated with activation for some GPCRs. We also observe conformational changes in the conserved PIF (P196[5.50] I114[3.40] and F260[6.44]) interaction, as well as the NPxxY (N322[7.49], P323[7.50], and Y326[7.53]) and DRY (D123[3.49], R124[3.50] and Y125[3.51]) motifs (Fig. 3d, e). In the active $\alpha_{1A}AR$, R124[3.50] forms hydrogen bond networks with Y204[5.58], Y326[7.53] and C329[7.56] (Fig. 3e). These structural changes allow the C-terminal helix ($\alpha$5 helix) of G$\alpha$ to engage the receptor core, as described below. In the extracellular view, due to the Nb29 binding, the conformation of ECL2, TMs 4 and 7 in the active state is closer to the receptor core, as discussed later in detail.

## Orthosteric ligand-binding pocket of $\alpha_{1A}AR$

Extensive site-directed mutagenesis studies have identified amino acids that form the binding pocket of the $\alpha_{1A}AR$, including residues responsible for subtype selectivity[34–42]. Our structures largely confirm these observations. $\alpha_{1A}AR$ structures bound to the endogenous agonist noradrenaline, selective partial agonist oxymetazoline, and selective antagonist tamsulosin are shown in Fig. 4. All three ligands form polar interactions with D106[3.32] which is a highly conserved residue involved in ligand binding in all aminergic receptors (Fig. 4 and Supplementary Fig. 7). The binding pocket of noradrenaline is formed by residues in TMs 3, 5, 6, and 7 (Fig. 4a, e). The noradrenaline has two catechol hydroxyl groups. The para-hydroxyl forms a hydrogen bond with S188[5.42], whereas meta-hydroxyl does not form polar interaction but is close to M292[6.55], a unique residue among ARs (Supplementary Fig. 7). Previous mutagenesis[34–36] and [$^{13}$C$^{\varepsilon}$H]methionine labeling NMR studies[37] support the role of S188[5.42] and M292[6.55] in ligand binding. The chiral $\beta$-hydroxyl forms a hydrogen bond with D106[3.32] and the amino group of the noradrenaline forms cation·$\pi$ stacking with the phenyl ring of F312[7.39] and a hydrogen bond with the backbone carbonyl of F312[7.39]. Noradrenaline forms extensive nonpolar interactions with highly conserved aromatic residues among ARs, including Y184[5.38], F288[6.51], and F289[6.52].

The oxymetazoline binds in a similar site (Fig. 4b, f), but the para-catechol hydroxyl is replaced by tertiary butyl, leading to the lack of polar interaction with S188[5.42], which may account for its partial agonism in receptor activation[12,14,15]. Tertiary-butyl group forms

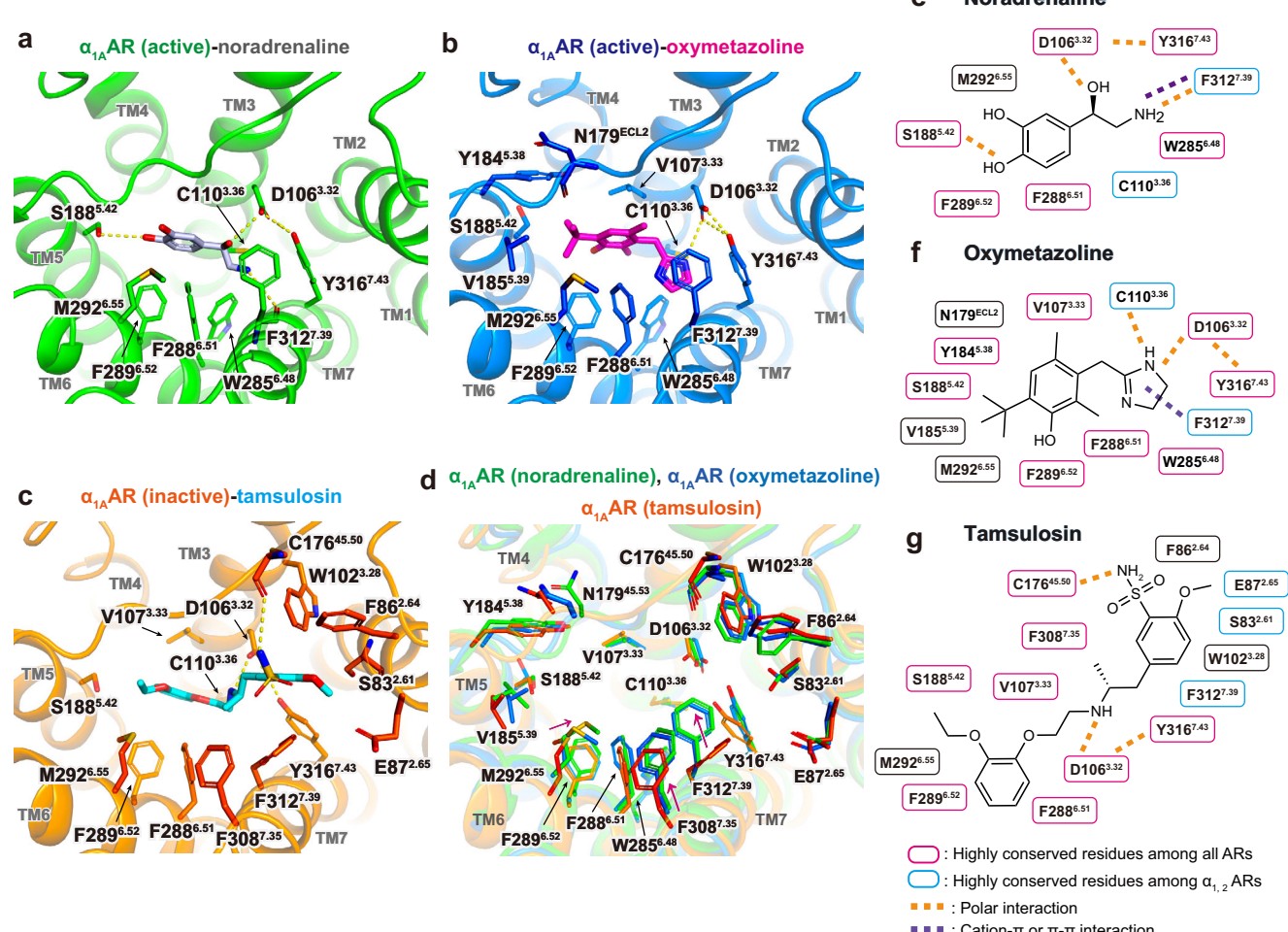

**Fig. 4 | Orthosteric ligand-binding pocket of α₁ₐAR.** Side view of α₁ₐAR ligand-binding site of noradrenaline (**a**), oxymetazoline (**b**), and tamsulosin (**c**). Residues within 4.1 Å distance of the ligands are shown in stick representation. Hydrogen bonds are shown as yellow dashed lines. **d** Comparison of residues involved in ligand-binding pockets. **e**–**g** Diagram of ligand interactions between α₁ₐAR and the ligands. Residues within 4.1 Å distance of the ligands are shown.

extensive van der Waals interaction with Y184⁵·³⁸, V185⁵·³⁹, S188⁵·⁴², M292⁶·⁵⁵ and the backbone carbonyl of N179^ECL2. Position 5.39 is a valine in α₁ₐAR, α₂ₐAR, and β₃AR, but is an alanine/isoleucine in other subtypes of ARs (Supplementary Fig. 7). A previous mutagenesis study showed that V185A⁵·³⁹ and M292L⁶·⁵⁵ mutations resulted in decreased oxymetazoline binding but not noradrenaline binding, and the equivalent A204V⁵·³⁹ and L312M⁶·⁵⁵ mutants of α₁ʙAR increased agonist binding[35,36]. In place of the chiral β-hydroxyl and the amino group of noradrenaline, oxymetazoline has an imidazoline ring which forms polar interactions with D106³·³² and C110³·³⁶, π-π stacking with F312⁷·³⁹, and aromatic interactions with W285⁶·⁴⁸, F288⁶·⁵¹ and Y316⁷·⁴². In α₁- and α₂ARs, C110³·³⁶ and F312⁷·³⁹ are conserved residues and involved in ligand recognition for imidazoline-type agonists[14–16]. Consistent with this result, a mutagenesis study indicated that F312A⁷·³⁹ and F312N⁷·³⁹ (the equivalent residues for βARs) mutations decreased oxymetazoline binding but not adrenaline binding[38]. This report also demonstrated that F308⁷·³⁵, a residue above the F312⁷·³⁹, influences oxymetazoline binding, even though it is too far away for a direct interaction[38].

The antagonist tamsulosin has two aromatic groups on each side of an ethyl-aminopropyl backbone (Fig. 4c, g). Similar to agonist binding, the ethylamine group of tamsulosin forms a hydrogen bond with D106³·³². The ethoxyphenoxy group forms van der Waals interaction with V107³·³³, S188⁵·⁴², M292⁶·⁵⁵, F288⁶·⁵¹ and F289⁶·⁵² (Supplementary Fig. 7). Unlike agonist binding, reorientation of F312⁷·³⁹ enlarges the binding pocket and enables the antagonist to bind

towards the extracellular vestibule, which is also called an exosite or secondary binding pocket with less conserved residues compared to the orthosteric pockets[12,43] (Fig. 4d). The sulfonamide group forms a polar interaction with the backbone carbonyl of C176⁴⁵·⁵⁰ and a non-polar interaction with F308⁷·³⁵. The methoxybenzene group forms non-polar interactions with F86²·⁶⁴, E87²·⁶⁵, W102³·²⁸, F312⁷·³⁹ and the backbone carbonyl of S83²·⁶¹ (Fig. 4c and Supplementary Fig. 7). F86²·⁶⁴ is a unique residue to α₁ₐAR relative to other ARs and was previously identified as a determinant for the interaction of the α₁ₐAR with various antagonists including HEAT and prazosin[39–41] (Supplementary Fig. 1). In addition, another mutation study indicated that three non-conserved residues (Q177⁴⁵·⁵¹, I178⁴⁵·⁵², N179⁴⁵·⁵³) in ECL2 are responsible for the α₁ₐAR selectivity of phentolamine and WB4101 over α₁ʙAR[42], but are not involved in binding tamsulosin.

## Ligand recognition of adrenergic receptor subtypes

Although all adrenergic receptors are activated by endogenous adrenaline and noradrenaline, their binding pockets are not identical. Comparisons of the key residues of the noradrenaline binding pockets in α₁ₐAR, α₂ₐAR[14] and β₁AR[11] reveal similar but different mechanisms of noradrenaline recognition (Fig. 5a, b). Compared to M⁶·⁵⁵ in α₁ₐAR, Y⁶·⁵⁵ in α₂ₐAR (conserved in all α₂ARs) forms a hydrogen bond with the meta-hydroxyl of the catechol ring, while the para-hydroxyl is involved in hydrogen bonds with S⁵·⁴² in both α₁ₐAR and α₂ₐAR. The rotamer of S⁵·⁴² differs between α₁ₐAR and α₂ₐAR, leading a distinct difference in

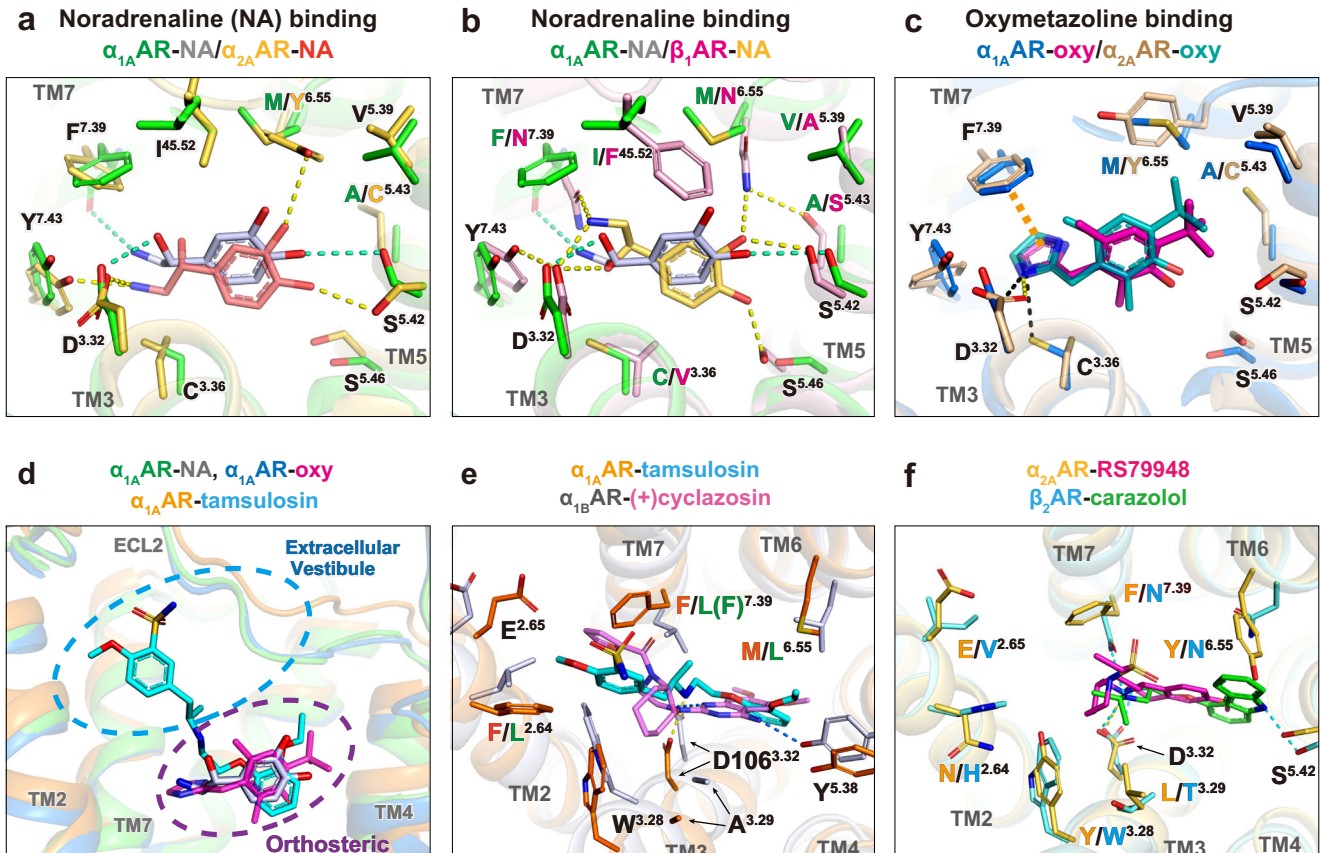

**Fig. 5 | Comparisons of ligand-binding pockets for the adrenergic receptors.**
**a** Noradrenaline binding between $\alpha_{1A}$AR (green sticks with green hydrogen bonds) and $\alpha_{2A}$AR (yellow sticks with yellow hydrogen bonds; PDB ID: 7EJ0).
**b** Noradrenaline binding between $\alpha_{1A}$AR (green sticks) and $\beta_1$AR (light pink with yellow hydrogen bonds; PDB ID: 7BU6). **c** Oxymetazoline binding between $\alpha_{1A}$AR (blue sticks with black hydrogen bonds) and $\alpha_{2A}$AR (wheat sticks with yellow hydrogen bonds; PDB ID: 7EJK). Aromatic interaction is shown as an orange dashed line. **d** Orthosteric and extracellular vestibule of $\alpha_{1A}$AR bound to noradrenaline

(gray) and tamsulosin (cyan). **e** Comparison of antagonists binding to $\alpha_{1A}$AR (orange sticks with yellow hydrogen bonds, tamsulosin is colored cyan), and $\alpha_{1B}$AR (gray sticks with blue hydrogen bonds and pink ligand; PDB ID: 7B6W). Note that $F^{7.39}$ of the $\alpha_{1B}$AR is mutated to $L^{7.39}$ for stabilization. **f** Comparison of antagonists binding between $\alpha_{2A}$AR (light orange sticks with yellow hydrogen bonds; ligand is a colored gray stick; PDB ID: 6KUX) and $\beta_2$AR (cyan sticks with cyan hydrogen bonds and green colored ligand; PDB ID: 2RH1).

the pose of the catechols. In $\alpha_{1A}$AR, the β-hydroxyl group interacts with $D^{3.32}$, and the amino group forms a hydrogen bond and a cation-π interaction with $F^{7.39}$ (Figs. 4a and 5a). In contrast, only the amino group forms hydrogen bonds with both $D^{3.32}$ and $Y^{7.43}$ in $\alpha_{2A}$AR. The noradrenaline binding pose of $\beta_1$AR is different from that of $\alpha_{1A}$AR (Fig. 5b). The meta-hydroxyl forms polar interaction networks with $S^{5.42}$, $S^{5.43}$ and $N^{6.55}$, and the para-hydroxyl forms a hydrogen bond with $S^{5.46}$ in $\beta_1$AR. Previous $\alpha_{1A}$AR mutagenesis studies indicated that double mutation of $S188A^{5.42}$ and $S192A^{5.46}$ decreased agonist binding rather than $S188A^{5.42}$ or $S192A^{5.46}$ alone[34]. Non-aromatic $N^{7.39}$ interacts with the amine group of noradrenaline through the polar interaction networks with $D^{3.32}$ and $Y^{7.43}$ in $\beta_1$AR. Moreover, the bulkier $F^{45.52}$ in ECL2 of $\beta_1$AR (conserved in all βARs) forms a non-polar interaction with noradrenaline[10,11].

We next compare the binding pose of imidazoline-type partial agonist oxymetazoline in $\alpha_{1A}$AR and $\alpha_{2A}$AR[14] (Fig. 5c). The oxymetazoline presents high selectivity for $\alpha_{1A}$AR and $\alpha_{2A}$AR over other αARs[4]. As mentioned before, oxymetazoline replaces the para-hydroxyl with the hydrophobic tertial-butyl group which no longer forms the polar interaction but engages in hydrophobic interactions with partially conserved $V^{5.39}$ in $\alpha_{1A}$AR. In $\alpha_{2A}$AR, oxymetazoline also does not form polar interaction with TM5, but $C^{5.43}$ is involved in the hydrophobic interaction. This position is $A189^{5.43}$ in $\alpha_{1A}$AR, while $S^{5.43}$ or $C^{5.43}$ in other ARs (Supplementary Fig. 7). In both $\alpha_{1A}$AR and $\alpha_{2A}$AR, the imidazoline ring is stabilized by π-π stacking with $F^{7.39}$ along with polar interaction with $D^{3.32}$, in contrast, $C^{3.36}$ (conserved in all αARs and V in βARs) forms a

weak polar interaction with the imidazoline ring only in $\alpha_{1A}$AR. Although most of the residues have the same orientation between noradrenaline and oxymetazoline binding in $\alpha_{1A}$AR, the orientation of $Y^{6.55}$ is shifted in $\alpha_{2A}$AR. This $Y^{6.55}$ is involved in $G_{i/o}$-biased signaling over β-arrestin recruitment for oxymetazoline among the other agonists such as noradrenaline, brimonidine and dexmedetomidine in $\alpha_{2A}$AR[14].

Compared to $\alpha_1$AR agonists, the $\alpha_1$AR antagonists have higher subtype-selectivity because they extend to the extracellular vestibule (Fig. 5d). As mentioned above, inactive $\alpha_{1A}$AR bound to tamsulosin reveals that unique ($F86^{2.64}$, and $M292^{6.55}$) and partially conserved ($S83^{2.61}$, $E87^{2.65}$, $W102^{3.28}$, $I178^{45.52}$, and $F312^{7.39}$) residues are involved in subtype selectivity (Fig. 3c, f and Supplementary Fig. 7). Recent crystal structure of inactive $\alpha_{1B}$AR bound to its selective inverse agonist (+)-cyclazosin, along with the chimeric $\alpha_{1B}$AR-$\alpha_{2C}$AR mutagenesis studies indicated that non-conserved residues $L^{2.64}$, $W^{3.28}$, $A^{3.29}$, $V^{45.52}$, and $L^{6.55}$ are important for the selectivity in $\alpha_{1B}$AR[18] (Supplementary Fig. 7). The (+)-cyclazosin is a derivative of prazosin in which piperazinyl quinazoline scaffold is introduced in a bulky cycloaliphatic group (Supplementary Fig. 1), leading to 100–1000 fold selectivity for $\alpha_1$ARs over $\alpha_2$ARs, and a slight preference for $\alpha_{1B}$AR over $\alpha_{1A}$AR. When comparing the $\alpha_{1A}$AR and $\alpha_{1B}$AR (Fig. 5e), both the tamsulosin and (+)-cyclazosin extend into the extracellular vestibule. Tamsulosin interacts with $F86^{2.64}$ more closely than the furan group of (+)-cyclazosin, which is consistent with the binding selectivity[44], however, a

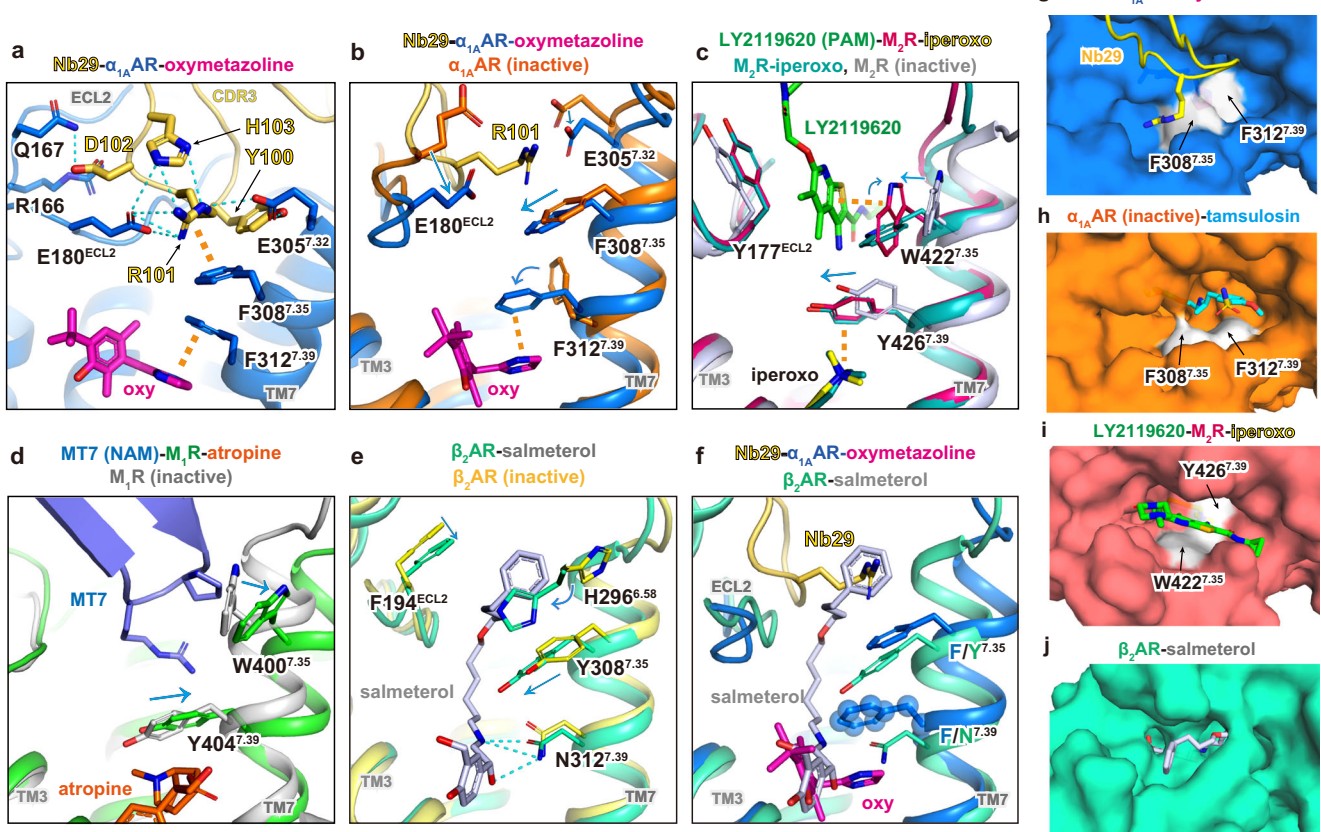

**Fig. 6 | Structural comparison of the Nb29 binding site.** Nb29 binding sites of (**a**) Nb29-$\alpha_{1A}$AR-oxymetazoline; **b** Nb29-$\alpha_{1A}$AR-oxymetazoline and inactive $\alpha_{1A}$AR; **c** LY2119620-M$_2$R-iperoxo (PDB ID: 4MQT), M$_2$R-iperoxo (PDB ID: 4MQS) and inactive M$_2$R (PDB ID: 3UON); **d** MT7-M$_1$R-atropine (PDB ID: 6WJC) and inactive M$_1$R (PDB ID: 5CXV); **e** $\beta_2$AR-salmeterol (PDB ID: 6MXT) and inactive $\beta_2$AR (PDB ID: 2RH1); **f** Nb29-$\alpha_{1A}$AR-oxymetazoline and $\beta_2$AR-salmeterol (PDB ID: 6MXT). Polar and aromatic interactions are shown as cyan and orange dashed lines, respectively. Conformational changes are shown with cyan arrows. **g–j** Surface representations of top views of Nb29-$\alpha_{1A}$AR-oxymetazoline, inactive $\alpha_{1A}$AR-tamsulosin, LY2119620-M$_2$R-iperoxo (PDB ID: 4MQT) and $\beta_2$AR-salmeterol (PDB ID: 6MXT).

stabilizing mutation of F$^{7.39}$ to L$^{7.39}$ in the $\alpha_{1B}$AR structure might affect the (+)cyclazosin binding mode[18]. In contrast to $\alpha_1$ARs, the antagonists of $\alpha_{2A}$AR and $\beta_2$AR do not interact with TM2 (Fig. 5e, f)[8,15]. The positions 2.64, 3.28, 3.29, and 45.52 are different from $\alpha_1$ARs and likely involved in the ligand selectivity. In addition, residues M$^{6.55}$ in $\alpha_{1A}$AR and N$^{6.55}$ in $\beta_2$AR allow antagonist interactions with the extracellular side of TM6, in contrast to the bulkier Y$^{6.55}$ in $\alpha_{2A}$AR.

It is known that the other $\alpha_{1A}$AR ligands such as A61603 (agonist) and silodosin (antagonist) have high selectivity for $\alpha_{1A}$AR (Supplementary Fig. 1)[37,44]. In these compounds, the phenyl rings corresponding to catechol have much bulkier substituents, suggesting that they may exhibit selectivity through interaction with $\alpha_{1A}$AR unique residues such as M292$^{6.55}$, A189$^{5.43}$ and the non-conserved residue V185$^{5.39}$.

### Structural insight into Nb29 binding

Nb29 binds to the extracellular side of $\alpha_{1A}$AR which is topologically distinct from the orthosteric agonist pocket (Fig. 2). This site has been shown to bind to allosteric modulators for muscarinic receptors[45–48]. We do observe a left shift of the agonist competition binding curves in the presence of Nb29 (Fig. 1c–e); however, these experiments are complicated by the fact that Nb29 is a competitive inhibitor of the radioligand [³H] prazosin. In cell signaling assays, Nb29 exhibits no agonist activity on its own, has no effect on EC$_{50}$ for oxymetazoline or noradrenaline, and slightly reduces the maximum efficacy of $\alpha_{1A}$AR activation (Supplementary Fig. 8a–d), suggesting that Nb29 appears to antagonize receptor activation or possibly block the ligand entry into

the orthosteric pocket. It should be noted that the radioligand competition assay was performed in equilibrium and the agonists had a longer incubation time to access the orthosteric pocket than in the signaling assay. In both assays, the effects of Nb29 are larger for oxymetazoline compared with noradrenaline, which is consistent with Nb29's binding selectivity towards the oxymetazoline-bound state of the $\alpha_{1A}$AR in the titration assay (Fig. 1b–e and Supplementary Fig. 8). Thus, Nb29 might be considered a weak positive allosteric modulator (PAM) or a neutral allosteric modulator, given that it binds to a known allosteric binding pocket in other GPCRs.

The Nb29 binding interactions are almost identical in oxymetazoline- and noradrenaline-bound Nb29-$\alpha_{1A}$AR complexes, but the map resolution is relatively poor in the noradrenaline-bound state (Fig. 1 and Supplementary Figs. 3 and 4). Thus, we used the oxymetazoline-bound state for structural analysis. Nanobodies consist of three complementarity-determining regions (CDRs). The relatively long CDR3 interacts with a broad range of residues from ECL2 and with E305$^{7.32}$ at the top of TM7 (Supplementary Figs. 9a–c and 10a, b). Among them, seven amino acids (R166, Q167, E171, T174, Q177, N179, and E305$^{7.32}$) are non-conserved residues in $\alpha_1$AR subtypes, suggesting that these residues are involved in nanobody specificity (Supplementary Figs. 7, 9a–c). Nb29 binding also stabilized the polar interaction network between ECL2 and R96$^{3.22}$, which is not observed in the inactive $\alpha_{1A}$AR structure without Nb29 (Supplementary Fig. 9d–f). Four residues of CDR3 (Y100, R101, D102 and H103) bind to the extracellular vestibule from the agonist-binding pocket (Fig. 6a). R101 of Nb29 forms charge networks with E180$^{ECL2}$ and E305$^{7.32}$, and a cation-$\pi$

interaction with F308[7.35] of $\alpha_{1A}$AR, stabilizing the inward conformation of TM7 (Fig. 6a, b). As mentioned above, F308[7.35] and the close-lid conformation of F[7.39] is important for the agonist binding in $\alpha$ARs (Fig. 4)[38], the π-π stacking of oxymetazoline with F312[7.39] might contribute to the Nb29 binding selectivity compared with the cation-π stacking of noradrenaline with F312[7.39].

The residues at position 7.35 have also been identified as critical residues for both PAM and negative allosteric modulator (NAM) binding of muscarinic acetylcholine receptors (MRs) by stabilizing the extracellular side of TM7 either in inward or outward conformations, respectively[45,46]. The LY2119620, a PAM for M$_2$R, binds to the extracellular vestibule and changes the conformation of W422[7.35] by an aromatic stacking, whereas those of the other residues are almost identical between the LY2119620-M$_2$R-iperoxo and M$_2$R-iperoxo complexes (Fig. 6c and Supplementary Fig. 10c, d)[45]. In contrast to this PAM, a peptide toxin MT7, the NAM for M$_1$R activation, stabilizes the outward displacement of TMs 6 and 7 through interactions with W400[7.35][46] (Fig. 6d and Supplementary Fig. 10e, f). Moreover, a mutagenesis study indicated that F330[7.35] in $\alpha_{1B}$AR is involved in the allosteric binding for conotoxin ρ-TIA, a selective NAM for $\alpha_{1B}$AR among $\alpha_1$ARs[49]. The Nb29 binding site also overlaps with the aryloxyalkyl tail of the selective agonist salmeterol binding in β$_2$AR (Fig. 6e, f)[12]. The smaller N[7.39] in β$_2$AR enables the salmeterol to extend into the extracellular vestibule to make aromatic interactions with F194[ECL2], H296[6.58] and Y308[7.35], which are unique to β$_2$AR (Supplementary Fig. 7). Among aminergic receptors, the aromatic amino acid at position 7.39 is observed in $\alpha$ARs (F[7.39]), muscarinic receptors (Y[7.39]) and histamine H$_3$ and H$_4$ receptors (F[7.39])[12,43,45,50]. The selective agonists targeting the extracellular vestibule of $\alpha$ARs are limited compared to other GPCRs such as βARs and muscarinic receptors[12,13,43,50] (Fig. 6g–j and Supplementary Fig. 1).

Subsequently, we compared the nanobody binding site with available class A GPCR structures in complex with extracellular nanobodies or antibody Fab fragments (Supplementary Fig. 10). Consistent with the review for GPCR antibodies[19], peptide-binding GPCRs are more frequently targeted by antibodies because they have relatively large binding pockets compared with the small-molecule binding GPCRs such as aminergic GPCRs. Only two structures in complex with extracellular nanobodies have been reported in class A GPCRs, which are the apelin receptor (APJ)[24] and the orexin receptor 2 (OX$_2$R)[51], although more structures of the intracellular binding nanobodies have been published for stabilizing the GPCR active conformations as G protein mimetics, such as Nb9-8 for M$_2$R[10,20,45] (Supplementary Fig. 10c–j). Anti-APJ nanobody JN241 antagonizes APJ through extensive interactions with extracellular loops of APJ and the insertion of CDR3 into the peptide-binding site[24], whereas anti-OX$_2$R nanobody Sb51 is positioned above the small-molecule agonist and partially overlaps with natural-peptide orexin B binding site[51]. In contrast to nanobodies, conventional antibodies and Fab fragments consist of heavy (CDRs H1-3) and light (CDRs L1-3) chains (Supplementary Fig. 10k–r). The antibody for protease-activated receptor 2 (PAR2) behaved as an antagonist by blocking ligand access from the extracellular region by both heavy and light chains (H1, H3, L2, and L3)[26]. In the angiotensin II type 2 receptor (AT2)[23], EP4[22], and D$_2$R structures[25], their antibodies allosterically enhance the ligand binding. The antibody for AT2 bound to the ECL1 and β-hairpin motif of ECL2 to stabilize the peptide agonist binding pocket, while the antibody for EP4 stabilizes the occluded β-hairpin of ECL2, leading to enhanced antagonist binding. In the D$_2$R-Fab3089 structure, the CDR-H2 stabilizes the antagonist spiperone which binds toward the extracellular vestibule.

Taken together, the Nb29 structure provides insights into allosteric binding and antibody recognition for $\alpha_{1A}$AR. Nb29 covers the extracellular surface of $\alpha_{1A}$AR like anti-APJ nanobody JN241 and anti-PAR2 Fab, whereas the CDR3 loop of Nb29 binds to a similar site of

PAM for M$_2$R (Fig. 6 and Supplementary Fig. 10). It should be noted that nanobodies are amenable to optimization due to the single variable domain. For example, the G protein-mimicking nanobody for β$_2$AR was optimized by directed evolution to increase its affinity to the receptor[10]; and the APJ nanobody antagonist JN241was rationally engineered into an APJ agonist by structure-guided site-directed mutation of CDR3[24].

## Selectivity of G protein interactions with adrenergic receptors

Noradrenaline- and oxymetazoline-bound $\alpha_{1A}$AR-miniGsq complexes are almost identical at the interfaces with miniGsq, as observed in $\alpha_{2A}$AR-Go complexes with different agonists[14]. Thus, we used the oxymetazoline-bound active state for structural analysis. Of the 15 residues at the carboxyl-terminal α5-helix of miniGsq (Fig. 7a)[29], seven residues are specific to Gq, including K[H5.12], L[H5.16], Q[H5.17], N[H5.19], E[H5.22], N[H5.24], and V[H5.26] (superscript, CGN G protein numbering system[52]); five residues [D[H5.13], I[H5.15], L[H5.20], Y[H5.23], and L[H5.25]] are conserved between Gs and Gq, and the other three residues [I[H5.14], M[H5.18] and R[H5.21]] are located on the opposite side of the interface. When comparing the interactions of $\alpha_{1A}$AR-miniGsq complexes with $\alpha_{2A}$AR-Go[14] and β$_2$AR-Gs complexes[9], the α5-helix of miniGsq is slightly shifted towards helix 8 of $\alpha_{1A}$AR (Fig. 7b–h). In the $\alpha_{1A}$AR-miniGsq complex, $\alpha_{1A}$AR forms six hydrogen bonds with the residues corresponding to Gq: between R213[5.67] and Q[H5.17], between the backbone carbonyl of G127[3.53] and N[H5.19], between T273[6.36] and the backbone carbonyl of N[H5.24]; the side chain of N[H5.24] forms hydrogen bond networks with the backbone carbonyl of C328[7.55], the backbone carbonyl of S330[8.47], and the side chain of Q331[8.48] (Fig. 7c, d). These residues of $\alpha_{1A}$AR are not conserved in $\alpha_2$ARs and βARs (Supplementary Fig. 11). The R[3.50] forms cation-π stacking interaction with Y[H5.23], which is also observed in the β$_2$AR-Gs complex, but not in $\alpha_{2A}$AR-Go complex as this position is C[H5.23] in Go protein (Fig. 7c–h). In the $\alpha_{2A}$AR-Go complex, polar interactions are observed between S[3.53] and N[H5.19], and hydrophobic interactions are predominant (Fig. 7e, f). In the β$_2$AR-Gs complex, one side of the α5-helix of the Gs protein forms a cluster of hydrogen-bond interactions with TM3 (I[3.54] and T[3.55]) and TM5 (E[5.64], Q[5.68], and K[5.71]), leading the α5-helix to shift towards TM5 (Fig. 7b, g, h). In addition to α5-helix, the N-terminal helix of G$_\alpha$ subunits are also involved in GPCR-G protein interactions[9,14]. The previous study indicates that polybasic cluster at the C terminus of M$_1$R, which is conserved among most G$_{q/11}$-coupling GPCRs, interacts with the G-protein G$\alpha_{11}$/β interface[53]. However, our structure lacks these regions and there are currently no other active structures of $\alpha_1$ARs. While $\alpha_{1A}$AR is predominantly coupled to G$_{q/11}$ proteins, a few studies have identified G$_{12/13}$[54] and β-arrestin signaling pathways[2,55]. Further studies will be required to better understand the mechanism of activation and selectivity of $\alpha_1$AR signaling.

## Discussion

Here, we present three cryo-EM structures of $\alpha_{1A}$AR in both active and inactive states. These structures reveal several structural aspects of $\alpha_{1A}$AR. The ligand-binding modes of the endogenous agonist noradrenaline and the imidazoline-type agonist oxymetazoline demonstrated a key aromatic interaction involving F312[7.39], which is conserved in $\alpha$ARs, and distinct ligand recognition by the unique residue M292[6.55]. The inactive $\alpha_{1A}$AR bound to tamsulosin reveals the subtype selectivity of antagonist binding pockets involving F86[2.64]. Our results also provide structural insights into nanobody recognition for $\alpha_{1A}$AR. Nb29 binds to the extracellular vestibule of $\alpha_{1A}$AR and the cationic residue of CDR3 stabilizes F308[7.35] which is the equivalent binding site of the positive allosteric modulator for M$_2$R. Finally, our active $\alpha_{1A}$AR structures provide insight into G protein binding selectivity by comparisons with $\alpha_{1A}$AR-miniGsq, $\alpha_{2A}$AR-Go, and β$_2$AR-Gs structures. Together with our $\alpha_{1A}$AR structures and previously published structures of $\alpha_2$ARs and βARs, the active and inactive structures of the major subtypes of the adrenergic receptor family have been

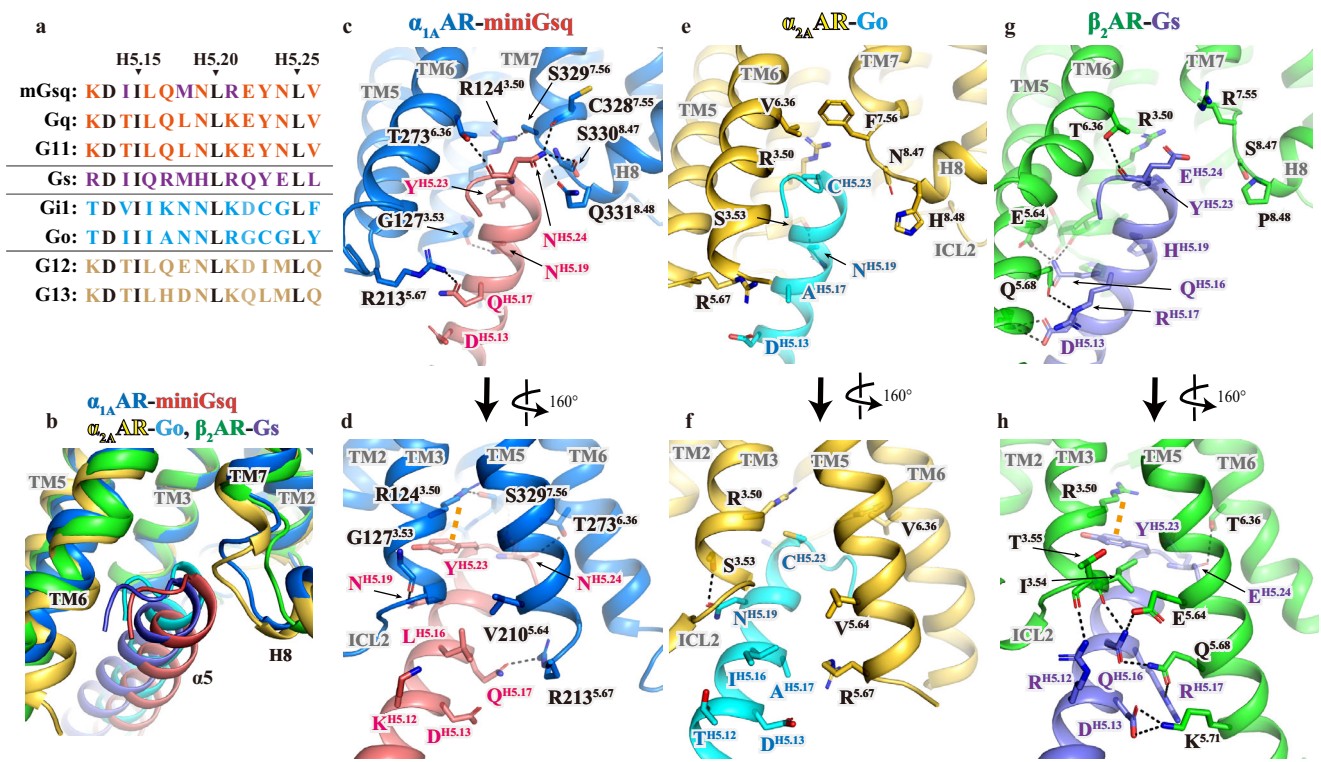

**Fig. 7 | Comparison of the receptor-G protein binding interfaces of the adrenergic receptor subtypes. a** Sequence alignments of the carboxyl-terminal α5-helix of G proteins subtypes. **b** Superimposition of the binding interfaces of α1AAR-miniGsq complexes with α2AAR-Go (PDB ID: 7EJ0) and β2AR -Gs (PDB ID: 3SN6) complexes. The receptors are used for alignment. The proteins are colored as follows: α1AAR (cobalt), miniGsq (pink), α2AAR (yellow), Go (cyan), β2AR (green), and Gs (blue purple). Detailed polar interactions and the equivalent residues of α1AAR-miniGsq (**c**, **d**), α2AAR-Go (**e**, **f**), and β2AR -Gs (**g**, **h**). The polar interactions are shown as black dashed lines.

determined. These results should facilitate the design of more selective and effective therapeutic drugs targeting both orthosteric and allosteric sites in this receptor family.

## Methods

### Construction

C-terminus truncated human α1AAR (residues 1–370, full length: 466) was modified by mutation of the N-linked glycosylation sites to glutamine (N7Q, N13Q and N22Q), N-terminal addition of the hemagglutinin signal peptide, FLAG-tag epitope, and C-terminal addition of the 8 × His-tag. For the active α1AAR structure, residues 223–261 of intracellular loop 3 were replaced with minimal T4L[28]. For the inactive α1AAR structure, we swapped residues from C205[5.59] to G275[6.38] with residues from T247[5.59] to L277[6.38] of kOR[21]. In addition, we introduced two thermostabilizing point mutations S113R[3.39] and M115W[3.41] which were previously used for structure determination of several inactive-state GPCRs[22,25,31]. The primers used in this study were obtained from Rui Biotech (Beijing, China), Xianghong Biotech (Beijing, China) or Genewiz (Beijing, China). The DNA sequencing analysis was performed at the Rui Biotech (Beijing, China).

### Expression and purification of α1AAR

Recombinant baculovirus was generated using the Bac-to-Bac Baculovirus Expression System (Thermo Fisher Scientific). *Sf9* insect cells at a cell density of 4 × 10⁶ cells/ml in ESF-921 insect media (Expressions Systems) with 20 μg/ml gentamycin and 1 μM ligand were infected with baculovirus and shaken at 27 °C for 2 days. Cells were harvested by centrifugation and stored at −80 °C until use.

To purify the protein, receptor-expressing *Sf9* cells were lysed by resuspension in a buffer containing 20 mM Tris-HCl pH 7.5, 1 mM EDTA, 10 μM ligand, 160 μg/ml benzamidine, 100 μg/ml leupeptin. The cell membranes were centrifuged at 10,000 g for 20 min at 4 °C. Receptor was extracted from cell membranes with solubilization buffer of 20 mM HEPES pH 7.5, 1% dodecyl maltoside (DDM), 0.03% cholesterol hemisuccinate (CHS), 0.2% Na cholate, 750 mM NaCl, and 30% glycerol. Iodoacetamide (2 mg ml⁻¹) was added to block reactive cysteines at this stage. Nickel-NTA agarose was added to the solubilized receptor without prior centrifugation. After stirring for 2 h at 4 °C, receptor-bound nickel resin was washed and poured into a glass column, and the receptor was eluted in 20 mM HEPES pH 7.5, 0.1% DDM, 0.03% CHS, 0.02% Na cholate, 750 mM NaCl, 10 μM ligand and 250 mM imidazole. Nickel resin-purified receptor was bound to M1-Flag affinity resin with 2 mM CaCl₂. Following extensive washing, detergent was gradually exchanged from DDM to 0.01% lauryl maltose neopentyl glycol (MNG). The receptor was eluted with 0.2 mg/ml Flag peptide and 5 mM EDTA and further purified by size exclusion chromatography (SEC) on a Sephadex S200 increase column (Cytiva) in a buffer of 20 mM HEPES pH 7.5, 0.01% MNG, 0.001% CHS, 100 mM NaCl and 10 μM ligand. The purified receptor was concentrated with a 50 kDa cutoff Amicon centrifugal filters (Millipore).

### Discovery of the conformationally selective α1AAR nanobody

The synthetic nanobody library displayed on the surface of BJ5465 yeast strain was obtained from Drs. A. C. Kruse (Harvard University) and A. Manglik (University of California San Francisco)[27]. The yeast cells were recovered in tryptophane dropout (-Trp) medium [prepared by Yeast Synthetic Drop-out Medium Supplements without tryptophane (sigma) and Yeast Nitrogen Base without amino acids (BD Difco) at pH 6.0] with 2% (w/w) glucose at 30 °C, and the nanobody was induced by -Trp medium with 2% (w/w) galactose at 25 °C. Expression levels of nanobody were estimated by staining with anti-HA antibody

(Cell Signaling Tech) and analyzing by flow cytometry with an Accuri C6 (BD Biosciences)

Induced yeast cells were washed and resuspended in a selection buffer (20 mM HEPES pH 7.5, 150 mM NaCl, 0.05% MNG, 0.005% CHS, 2.8 mM CaCl2, 0.1% (w/v) bovine serum albumin and 5 mM maltose). Nanobody clones against the purified FLAG-tagged $\alpha_{1A}AR$ (C-terminus truncated after residue 370) bound to oxymetazoline were enriched by two rounds of magnetic-activated cell sorting (MACS) and four rounds of fluorescence-activated cell sorting (FACS) (See Fig. 1a). For the first round of the MACS, $5 \times 10^9$ yeast cells were precleared by incubating with Alexa Fluor-647 conjugated anti-FLAG M1 antibody (M1-647, prepared by anti-FLAG M1 antibody and Alexa Fluor 647-NHS ester) and anti-Alexa Fluor-647 microbeads (Miltenyi) and passed LD column (Miltenyi) to remove nonspecific nanobody. Flowed-through yeast cells were washed with the selection buffer, then incubated with 0.2 μM $\alpha_{1A}AR$ bound to oxymetazoline, the antibody and the microbeads. After incubation at 4 °C for 30 min, yeast cells were loaded on the LD column, washed with the selection buffer and the eluted yeast cells ($3.4 \times 10^6$ cells) by plunger were expanded and used in a subsequent round of MACS. The second round of MACS was performed similarly to the first, but beginning with $4 \times 10^8$ yeast cells and using biotin-conjugated anti-FLAG M1 antibody Fab fragment (anti-FLAG M1 antibody was digested by papain then labeled with biotin-NHS ester), streptavidin microbeads (Miltenyi) and LS column (Miltenyi) were used, and $5 \times 10^6$ yeast cells were eluted.

Subsequently, we performed four rounds of FACS by FACSAria II (BD Biosciences) (See Supplementary Fig. 2). For the selection rounds 3 and 6, yeast cells were stained with Alexa Fluor-488 or −647 conjugated anti-HA antibody (Cell Signaling Tech) and 0.1 μM FLAG-tagged $\alpha_{1A}AR$ with anti-FLAG M1-647 or −488. For the selection rounds 4 and 5, in order to enrich for conformational selective nanobodies, yeast cells were stained with two different populations of $\alpha_{1A}AR$ labeled with anti-FLAG M1-488 and −647 fluorophores, one bound with oxymetazoline and another bound to tamsulosin. Staining yeast cells for each round of FACS experiments were the following; $5 \times 10^7$ cells for round 3 and $1 \times 10^7$ cells for rounds 4–6. After round 6, the sorted yeast cells were diluted and plated on -Trp agar plates. Single clones were sequenced and cloned into the periplasmic expression vector pET26b, containing an N-terminal pelB signal sequence and a C-terminal histidine tag, and transformed into BL21(DE3) *Escherichia coli*. Cells were induced in Terrific Broth medium with 2 mM MgCl2, 0.1% glucose and 50 μg/ml kanamycin at an OD600 of 0.7 with 1 mM IPTG and incubated with shaking at 25 °C for 20 h. Periplasmic protein was obtained by osmotic shock in a buffer containing 0.2 M Tris pH 8.0, 0.5 mM EDTA and 0.5 M sucrose at 4 °C for 1 h, then diluted 4 times and incubated for another one hour. The lysate was centrifuged and the supernatant was purified by Ni-NTA resin and size-exclusion chromatography.

For the on-yeast titration assay, Nb29-displayed yeast cells were stained with the Alexa Fluor-647 conjugated anti-HA antibody and several concentrations of purified $\alpha_{1A}AR$ fused at the C-terminus to an enhanced green fluorescent protein in the presence or absence of 500 μM ligands in the selection buffer. Yeast cells were analyzed by Accuri C6 and the ratio of double-positive yeast cells among anti-HA positive cells was calculated.

### Purification of the Nb29-$\alpha_{1A}AR$-miniGsq and the $\alpha_{1A}AR$-Nb6 complexes

We modified the expression and purification method of miniGsq from the previous report[29,56], in which miniGsq was used instead of miniGq for the expression in *Escherichia coli*. The pET21a plasmid encoding miniGsq and N-terminal histidine tag was transformed into BL21(DE3) *Escherichia coli*. Cells were induced in Terrific Broth medium at an OD600 of 0.6 with 1 mM IPTG and incubated with shaking at 25 °C for

20 h. Cells were harvested and lysed by sonication in a buffer containing 40 mM Hepes pH 7.5, 100 mM NaCl, 10 mM imidazole, 10% glycerol, 5 mM MgCl2, 50 μM guanosine diphosphate (GDP), 100 μM dithiothreitol, 160 μg/ml benzamidine, and 100 μg/ml leupeptin. The lysate was centrifuged at 10,000 g for 20 min at 4 °C and the supernatant was immobilized by Ni-NTA resin. The eluate was further purified by size-exclusion chromatography in a buffer containing 10 mM HEPES pH 7.5, 100 mM NaCl, 1 mM MgCl2, 10 μM GDP and 100 μM tris(2-carboxyethyl)phosphine (TCEP).

The Nb29-$\alpha_{1A}AR$-miniGsq complex was prepared by mixing with the purified $\alpha_{1A}AR$ bound the agonist (oxymetazoline or noradrenaline), Nb29 and miniGsq in a 1:1.2:1.2 molar ratio and supplemented with apyrase. After incubation for 2 h at room temperature, the mixtures were purified by size exclusion in SEC buffer containing 20 mM HEPES pH 7.5, 100 mM NaCl, 0.002% MNG, 0.0002% CHS, 2 mM MgCl2, 10 μM agonist and 100 μM TCEP. Fractions containing the complex were concentrated to 5–10 mg/ml using a 50 kDa molecular weight cutoff Amicon Ultra concentrator. For the $\alpha_{1A}AR$-Nb6 complexes, Nb6 from gene synthesis[33] was prepared as the same as nanobody purification described above. Purified $\alpha_{1A}AR$-kOR swap construct and Nb6 were mixed in a 1:1.5 molar ratio in the presence of tamsulosin and incubated on ice for 30 min. The mixture was further purified by size exclusion in SEC buffer containing 20 mM HEPES pH 7.5, 100 mM NaCl, 0.002% MNG, 0.0002% CHS, 2 mM MgCl2, and 10 μM ligand. Fractions containing the complex were concentrated using a 50 kDa molecular weight cutoff Amicon Ultra concentrator.

### Ligand binding assay

Ligand binding assays were performed with *Sf9* cell membrane or purified $\alpha_{1A}AR$-bound M1-Flag affinity resin. Receptor-expressing cells were harvested and homogenized in a binding buffer containing 20 mM Tris−HCl (pH 7.5) and 100 mM NaCl. After centrifugation, the pellet was homogenized in binding buffer and used as the membrane fraction in binding assays. The purified $\alpha_{1A}AR$-bound M1-Flag affinity resin was resuspended in binding buffer containing 20 mM HEPES pH 7.5, 100 mM NaCl, 0.01% MNG, 0.001% CHS and 2 mM CaCl2.

Radioligand binding assay was performed using [³H]prazosin (PerkinElmer). Receptors were incubated for 1 h at room temperature with various concentrations of ligand in a total volume of 100 μl. After the reaction, the mixture was trapped on Whatman GF/B glass filters. Bound and free radioligands were separated by washing with ice-cold binding buffer. Radioactivity was measured on a MicroBeta2 liquid scintillation counter (PerkinElmer). All binding assay measurements were analyzed using the Prism 9 software (GraphPad).

### Cryo-EM sample preparation and data acquisition

Two of the Nb29−$\alpha_{1A}AR$-miniGsq complexes were collected cryo-EM data at Tsinghua University, and the $\alpha_{1A}AR$-Nb6 complex was performed data acquisition by the Shuimu Bioscience (Beijing). The purified protein complexes were concentrated to 5–10 mg/mL. 4 μl of sample were applied to the glow-discharged holey carbon grids (Au R1.2/1.3, 300 mesh) purchased from Quantifoil for the Nb29-$\alpha_{1A}AR$-miniGsq complexes, and from Zhongjingkeyi Technology (Beijing) for the $\alpha_{1A}AR$-Nb6 complex. The grids were blotted for 3.0 s and flash-frozen in liquid ethane cooled by liquid nitrogen with Vitrobot (Mark IV, Thermo Fisher Scientific) before being transferred to a 300 kV Titan Krios microscope equipped with Gatan K3 Summit detector and a GIF Quantum energy filter (slit width 20 eV) or Falcon-4 detector and no energy filter. AutoEMation was used for the fully automated data collection in Tsinghua University[57]. The total dose of each stack was about 50 e⁻/Å². All frames in each stack were aligned and summed using the whole-image motion correction program MotionCor2[58] and binned to a pixel size of 1.083 Å/1.098 Å/0.860 Å for Nb29-$\alpha_{1A}AR$-miniGsq bound to oxymetazoline/ Nb29-$\alpha_{1A}AR$-miniGsq bound to noradrenaline/ $\alpha_{1A}AR$-Nb6 bound to tamsulosin datasets, respectively. The defocus

value of each image, which was set from −1.3 to −1.8 μm during data collection, was determined by Gctf[59].

## Cryo-EM data processing

For Nb29-α$_{1A}$AR-miniGsq bound to oxymetazoline/ Nb29-α$_{1A}$AR-miniGsq bound to noradrenaline/ α$_{1A}$AR-Nb6 bound to tamsulosin datasets, 700/1099/1911 dose-weighted micrographs were imported into cryoSPARC and CTF parameters were estimated by using patch-CTF, respectively. 1,161,201/2,597,118/2,334,500 particles picked by blob picker or template picker were extracted and subjected to 2D classification. 697,792/774,350/885,066 particles remained to generate the initial model by Ab-Initio Reconstruction and perform the following iterative rounds of heterogeneous refinement. After non-uniform refinement and local refinement, 359,833/ 393,438/ 285,284 particles yield the maps which reached the resolutions at 2.92 Å/ 3.52 Å/3.35 Å.

## Model building and refinement

The atomic coordinate of the Nb29-α$_{1A}$AR-miniGsq and the α$_{1A}$AR-Nb6 complexes was generated by combining homology modeling and de novo model building. An initial structure model for the active α$_{1A}$AR was predicted by the homology model from GPCRdb (gpcrdb.org)[60], α$_{1A}$AR-kOR was generated by AlphaFold2[61] and the structure model of Nb29 was predicted by the homology model from swiss-model[62]. The initial models of miniGsq and Nb6 were imported from miniGs (PDB code: 5G53)[29] and Nb6 (PDB code: 6VI4)[33], respectively. The cryo-EM model was docked into the electron microscopy density map using UCSF Chimera[63], followed by iterative manual adjustment and rebuilding in PHENIX[64] and COOT[65]. The structures were refined against the corresponding map using PHENIX and COOT in real space with secondary structure and geometry restraints. Figures were created using the PyMOL Molecular Graphics System v.2.4.0 (Schrödinger, LLC), UCSF Chimera and the UCSF Chimera X1.3 package.

## Glo-sensor signaling assay

To determine the signaling profile of Nb29 against α$_{1A}$AR, we used a cAMP Glo-sensor kit (Promega) with an engineered Gsq protein in which 15 residues at the C-terminus of Gs protein were replaced with those of Gq protein. The Gsq protein could be activated by the α$_{1A}$AR and stimulates intracellular cAMP production. In brief, pGlo-Sensor™-22F plasmid, receptor plasmid, and Gsq plasmid were transfected into HEK293T cells. At 24 h after transfection, the cells were switched into a $CO_2$-Independent Medium (Gibco) and incubated with GloSensor™ cAMP reagent. The mixture was then transferred to a 96-well white plate. The 96-well plate is placed at 37 °C in the dark for 1 h, then placed at room temperature in the dark for 1 h before use. The luminescence signal was measured by Ensight™ plate reader (PerkinElmer) around 10–15 min after the addition of the agonist and/or Nb29. The result curves were calculated and fitted by GraphPad Prism 9.

## Reporting summary

Further information on research design is available in the Nature Portfolio Reporting Summary linked to this article.

## Data availability

Atomic coordinates and cryo-EM maps for the reported structures were deposited in the Protein Data Bank under accession codes 7YM8 (Nb29-α$_{1A}$AR-miniGsq bound to oxymetazoline), 7YMH (Nb29-α$_{1A}$AR-miniGsq bound to noradrenaline), and 7YMJ (α$_{1A}$AR-Nb6 bound to tamsulosin), and in the Electron Microscopy Data Bank under accession codes EMDB-33924 (Nb29-α$_{1A}$AR-miniGsq bound to oxymetazoline), EMDB-33928 (Nb29-α$_{1A}$AR-miniGsq bound to noradrenaline), and EMDB-33930 (α$_{1A}$AR-Nb6 bound to tamsulosin), respectively. Previously published structures can be accessed via accession codes:

5G53, 6VI4, 7EJ0, 7BU6, 7EJK, 7B6W, 2RH1, 6KUX, 4MQT, 3UON, 6WJC, 5CXV, 6MXT, 7EJ0, 7UL2, 6WJC, 6KNM, 7L1V, 5YWY, 7DFP. Source data are provided with this paper.

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

## Acknowledgements

We are grateful to the Tsinghua University Branch of China National Center for Protein Sciences (Beijing) and the Cryo-EM core of the Shuimu Bioscience (Beijing) for providing the cryo-EM and computation support; to Drs. A. C. Kruse (Harvard University) and A. Manglik (University of California San Francisco) for supplying the synthetic nanobody library; to De Li from Center of Biomedical Analysis of Tsinghua University for her assistance on radioligand binding assay. This work was supported by Beijing Frontier Research Center for Biological Structure, Beijing Advanced Innovation Center for Structural Biology, TOYOBO Biotechnology Foundation fellowship (Y.T.), Yamada Science Foundation (Y.T.), China Postdoctoral Science Foundation (2019T120110 to Y.T.), the National Natural Science Foundation of China (Grants 32122041 to X.L. and 9216920007 to L.Z.), the National Key R&D Program of China (2020YFA0509300 to C.Y.), Beijing Nova Program (Z201100006820039 to C.Y.), and Start-up funds from Tsinghua-Peking Center for Life Sciences and Tsinghua University (C.Y. and X.L.).

## Author contributions

Y.T. performed construction, protein expression, purification, and characterization of nanobodies; A.Z., F.K., J.Z., N.W., and Y.T. obtained cryo-EM data and processed the cryo-EM data under the supervision of X.L. and C.Y., Y.T. and S.S. performed nanobody screening under the supervision of L.Z., Y.T. modeled and refined the structures from cryo-EM density maps; Y.T., X.L., and B.K.K. wrote the manuscript; All authors discussed the results and commented on the manuscript; X.L. and X.S. managed the lab experiments; B.K.K., X.L., and Y.T. supervised the overall project.

## Competing interests

B.K.K. is a co-founder of and consultant for ConfometRx, Inc. The other authors declare no competing interests.
