## [Peer Review File · Nature Communications]

Structural basis of α 1A-adrenergic receptor activation and recognition by an extracellular nanobodyREVIEWER COMMENTS

Reviewer #1 (Remarks to the Author):

The authors have determined the structure of the alpha1A-AR in both active and inactive states using norepinephrine and the alpha1A-AR selective agonist, oxymetazoline, and somewhat alpha1A-AR selective antagonist, Tamsulosin. It is a solid work. Some of the aspects of the structure as it pertains to agonism and antagonism have also been previously identified by mutagenesis. Insights into selectivity in G-protein coupling has provided novel information but not a lot of mechanistic insight. A novel nanobody is suggested to be a PAM and the structure offers insight into that binding mode, but PAM characterization is incomplete.

Major comments:

1. It would help if the authors summarized some of the more unexpected binding modes as I struggled to see much that have not been previously identified through mutagenesis. This would increase the impact and the ease of reading your manuscript. While I love structure and agree it is more definitive than mutagenesis, much of the key residues involved in selective and non-selective binding and activation have been suggested with prior mutagenesis. I think this is nice that one confirms the other.
2. Line 169-170 and 214. This residue (Met 292) was previously identified as alpha1A-AR agonist specific in "Hwa J, Graham RM, Perez DM. Identification of critical determinants of alpha 1-adrenergic receptor subtype selective agonist binding. J Biol Chem. 1995 Sep 29;270(39):23189-95" which was not included in your manuscript. As this is one of the major findings of your work and may alter the orientation of imidazolines in your structure, it should be included and commented on in your discussion. This residue rat alpha1A-AR Met 293, equivalent to your human alpha1A-AR Met 292, is a key residue identified for selective alpha1A-AR agonism. Conversion of the alpha1B-AR leucine314 to equivalent in the rat alpha1A (met 292) resulted in increased affinity for imidazolines (cirazoline, oxymetazoline) and the alpha1A selective phenethylamine agonist methoxamine but not for non-alpha1A selective agonists, such as NE or Epi. The data was interpreted that this residue interacted with agonists that contain ortho-substituents that all alpha1A -AR selective agonists have. If this interpretation is correct, the imidazoline phenyl group should be rotated in your model to account for these ortho-interactions with Met 292. This may be possible in your model as the phenyl substituents were not assigned interactions in your structure. This might also be accommodated in your model as it would not disturb the tert-butyl interactions. Please add this reference and comment on this as this is proposed as a key residue identified in your work (line 381).
3. There are multiple issues with characterization of NB29 as a PAM. I think that these issues should be resolved before you can say it is a PAM. Otherwise, you just describe NB29 binding site.

- a. As it seems that all prazosin-like antagonists also bind in the exosite and ECL2 (Deluigi et al, 2022 Nature Commun.), NB29 may have effects on 3H-prazosin binding that are not allosteric. Verification of Increased affinity by ligand binding should be performed with 125I-HEAT
- b. Was ligand binding performed at equilibrium? Was that even determined? Allosteric binding needs to be done at 4x equilibrium as artifacts appear and your binding differences are only 3- fold apart.
- c. All the binding curves are normalized to their own binding max which may produce leftward shifts if Nb29 masks oxymetazoline binding or you are not at equilibrium. The binding curves should be normalized not to their own maximum but to the maximum binding (100%) of alpha1A-AR with oxymetazoline.
- d. A 3-fold shift in affinity is not very large to suggest a PAM. To confirm PAM effect would need to determine effects on signaling or other downstream effector.
- e. Is Nb29 + oxymetazoline binding effects specific to oxymetazoline or are similar effects seen with other ligands? Need to see effects with endogenous ligands also.
- f. 3-fold differences call for more points on the curve. It appears that there are no error bars on the supplemental Fig 1d for alpha1A + Nb29. Is that a mistake or there really is no error in N=3 experiments?

Minor Comments:

1. Why was oxymetazoline used for the structure as opposed to the better selective imidazoline agonists, such as cirazoline or A61603? Oxymetazoline is a very poor agonist at the alpha1A-AR. Was other agonists attempted but oxymetazoline gave the better crystal?
2. In fig 4e, it appears that TM3 of the alpha1B-AR structure has a more inward tilt than in the alpha1A-AR. Please comment. Is it consequential?
3. Line 48- I am unaware of any alpha1A/D designation being generally accepted or published at the time. There was an alpha1A/C designation that Paul Simpson published several manuscripts with that designation. The past nomenclature is confusing enough for the non-adrenergic scientists, maybe best to leave that out (alpha1A/D)?
4. S188 was identified as responsible for alpha1A-AR agonism first in this publication, "Hwa J, Perez DM. The unique nature of the serine interactions for alpha 1-adrenergic receptor agonist binding and activation. J Biol Chem. 1996 Mar 15;271(11):6322-7", which you do not reference. Although this publication suggested that the meta-hydroxyl interacted with S188 because phenylephrine, a full agonist, only contains a meta-hydroxyl, the main point of this publication was that S188 was responsible for agonism which is confirmed in your structure.
5. Line 227-8. The "Hwa J, Perez DM. The unique nature of the serine interactions for alpha 1-adrenergic receptor agonist binding and activation. J Biol Chem. 1996 Mar 15;271(11):6322-7", publication also first mentions the differences in the catechol ring orientation (about 120 degrees) between alpha1 and beta-

ARs. I couldn't tell the degree difference in your figures for the catechol ring, but please ref this publication.

6. Imidazolines have been identified to bias signaling towards Gs for the alpha1A-AR (Evans et al., *molpharm* 79: 298, 2011; da Silva et al., *Br. J. Pharmacol*, 174: 2318, 2017). Is there any indication for this in your model?

Reviewer #2 (Remarks to the Author):

Toyoda et al describe three new cryo-EM structures of the alpha1a-adrenergic receptor (alpha1AR), two coupled to a G protein in an active conformation and one in the inactive state. The two active state structures are bound either to the native agonist noradrenaline or to the selective agonist oxymetazoline. The inactive state structure was possible only through a combination of two stabilising mutations, engineering the receptor to bind Nb6 on the intracellular surface and binding of a newly selected nanobody to the extracellular surface (Nb29). The structures are of sufficient quality to substantiate the claims made by the authors in the manuscript on ligand binding and the activation mechanism.

The manuscript describes in great detail each of the three structures, how the ligands bind and how they differ from other alpha and beta ARs. The activation mechanism of the receptors is directly analogous to other Class A receptors. The structures provide a basis for understanding ligand selectivity between the different receptors. In addition, Nb29 is an interesting addition to the range of nanobodies identified to date that bind to the extracellular surface of GPCRs as it stabilises the conformation of the inactive state and therefore behaves like a PAM.

The manuscript is well written and presented and I have only a few minor comments that need addressing.

1. Line 96, Supplementary Figs 2b and 2c are not mentioned in the main text.
2. Line 118, insert '...had been determined' after 'complex'?
3. Line 144, state between which atoms the distance of 14.5Å was measured.
4. Line 224, It is mentioned that the para-hydroxyl of the catechol hydrogen bonds to Ser5.42 in both alpha1AR and alpha2AR, suggesting the mode of catechol binding is similar. However, because the rotamer of Ser5.42 differs between the two receptors, then there is a distinct difference in the pose of the catechols. Perhaps this would be good to mention.

5. Line 227, delete 'more'
6. Line 327, define 'ligand' as agonist or antagonist
7. Line 328, the words 'diagonally' and 'upright' are meaningless without a context. Reformulate.
8. Lines 357 and 387, you cannot say 'may be involved in the G protein coupling selectivity' as you have provided insufficient data directly pertinent to the role of the specific residues in this context e.g. mutagenesis data of specific residues that alter the coupling specificity. Please delete.
9. Line 700, define unmodelled grey density in Figs 1a,b and c
10. Fig 4b. The hydrogen bonds between noradrenaline and the side chains of alpha1AR are missing. Please add.
11. Supplementary Figs 1, 2 and 4. Please define in the legends the number of technical and biological replicates for the binding data (not just 'n=3 measurements' which is ambiguous). Also give the errors for all the K_i values and define what they are.
12. Supplementary Fig 3, and Fig 5g,h, and Fig 1b,d,f. Give sigma values and cut-offs for the density mesh, and which program was used.
13. Supplementary Fig 6. There is no panel 'b', so it seems superfluous to label this 'a'. What do the red amino acid residues mean compared to the black (also in Sup Fig 9)?
14. Supplementary Fig 7. What do the triangles mean in panel b?

Reviewer #3 (Remarks to the Author):

The manuscript by Toyoda et al presents the cryoEM structures of the human GPCR alpha-1a adrenergic receptor that responds to adrenaline and noradrenaline and is involved in smooth muscle contraction and cognitive function. Three conformations were solved. First, the active conformation in complex with the miniGq protein bound either to the endogenous agonist (noradrenaline) or a selective agonist of the alpha-1a receptor (oxymetazoline) and stabilized by the nanobody (VHH) Nb29 that binds to the extracellular part of the receptor. The third structure reports the inactive conformation of the alpha-1a adrenergic receptor in complex with another nanobody (Nb6) that binds to the intracellular part of the receptor.

The results are original since in the alpha-1 receptors family only the crystal structure of the alpha-1b receptor was previously reported (Deluigi et al (2022) Nat Comm). The selectivity of ligand binding was largely reported in the study.

In addition, in the present work the author discovered the Nb29 nanobody as a positive allosteric modulator that specifically binds to the extracellular vestibule of the alpha-1a receptor.

Comments:

- Do the nanobodies Nb29 and Nb6 are required to stabilize the active and inactive conformations, respectively, of the alpha-1a receptor?
- Evidence for positive allosteric modulation of the Nb29 is shown only by binding experiments. It would be even more compelling if functional data using signaling assays were provided. Does the Nb29 alone (absence of alpha-1 receptor agonist) could activate the receptor?
- What is the affinity of the nanobody Nb29?
- Which alpha-1 receptor constructs are using for nanobody screening (line 463)? It should be clarified.
- Line 76: "class A" should be added since the structure of a nanobody in complex with the metabotropic glutamate mGlu2 receptor was reported by the Beili Wu's group (Lin et al (2021) Nature).
- Line 296: repetition of line 283.

Reviewer #1 (Remarks to the Author):

The authors have determined the structure of the alpha1A-AR in both active and inactive states using norepinephrine and the alpha1A-AR selective agonist, oxymetazoline, and somewhat alpha1A-AR selective antagonist, Tamsulosin. It is a solid work. Some of the aspects of the structure as it pertains to agonism and antagonism have also been previously identified by mutagenesis. Insights into selectivity in G-protein coupling has provided novel information but not a lot of mechanistic insight. A novel nanobody is suggested to be a PAM and the structure offers insight into that binding mode, but PAM characterization is incomplete.

We thank the reviewer for the positive comments on our work and we appreciate the suggestions to help us improve the manuscript. The point-by-point responses to the concerns/suggestions are provided below.

Major comments:

1. It would help if the authors summarized some of the more unexpected binding modes as I struggled to see much that have not been previously identified through mutagenesis. This would increase the impact and the ease of reading your manuscript. While I love structure and agree it is more definitive than mutagenesis, much of the key residues involved in selective and non-selective binding and activation have been suggested with prior mutagenesis. I think this is nice that one confirms the other.

The previous mutagenesis studies are thoroughly performed and our structures are consistent with previous works. We added the sentences regarding this in line 170.

2. Line 169-170 and 214. This residue (Met 292) was previously identified as alpha1A-AR agonist specific in “Hwa J, Graham RM, Perez DM. Identification of critical determinants of alpha 1-adrenergic receptor subtype selective agonist binding. J Biol Chem. 1995 Sep 29;270(39):23189-95” which was not included in your manuscript. As this is one of the major findings of your work and may alter the orientation of imidazolines in your structure, it should be included and commented on in your discussion. This residue rat alpha1A-AR Met 293, equivalent to your human alpha1A-AR Met 292, is a key residue identified for selective alpha1A-AR agonism. Conversion of the alpha1B-AR leucine314 to equivalent in the rat alpha1A (met 292) resulted in increased affinity for imidazolines (cirazoline, oxymetazoline) and the alpha1A selective phenethylamine agonist methoxamine but not for non-alpha1A selective agonists, such as NE or Epi. The data was interpreted that this residue interacted with agonists that contain ortho-substituents that all alpha1A-AR selective agonists have. If this interpretation is correct, the imidazoline phenyl group should be rotated in your model to account for these ortho-interactions with Met 292. This may be possible in your model as the phenyl substituents were not assigned interactions in your structure. This might also be accommodated in your model as it would not disturb the tert-butyl interactions. Please add this reference and comment on this as this is proposed as a key residue identified in your work (line 381).

We added this reference in lines 172 and 181 (ref 35) and mentioned this mutation. According to the structure comparisons with the other adrenergic receptor structures (Figs 3 and 4), the region surrounded by unique residues such as M292^{6,55}, A189^{5,43}, and non-conserved residue V185^{5,39} is important for the subtype selectivity of α_{1A} AR.

We added a sentence about A189^{5,43} in line 242 and moved one paragraph at lines 272-276 from the former section in the previous manuscript (line 211).

According to the map, the phenyl group of oxymetazoline has the correct orientation (Fig 3 and Supplementary Fig. 3).

3. There are multiple issues with characterization of NB29 as a PAM. I think that these issues should be resolved before you can say it is a PAM. Otherwise, you just describe NB29 binding site.

We thank the reviewer for the suggestion. We characterized the function of Nb29 in cell signaling assay and found that Nb29 exhibits no agonist activity on its own, while it also reduces the maximum efficacy of α_{1A} AR activation. Most likely Nb29 blocks the entry of oxymetazoline to the orthosteric pocket. Following the reviewer’s suggestion, we are more conservative in our definition of Nb29 as a PAM in the revised manuscript. See the following paragraph

“Nb29 binds to the extracellular side of α_{1A} AR which is topologically distinct from the orthosteric agonist pocket (Fig. 1). This site has been shown to bind to allosteric modulators for muscarinic receptors (ref. 45-48). We do observe a left shift of the agonist competition binding curves in the presence of Nb29 (Supplementary Figs. 1d-f); however, these experiments are complicated by the fact that Nb29 is a

competitive inhibitor of the radioligand [³H] prazosin. In cell signaling assays, Nb29 exhibits no agonist activity on its own, has no effect on EC50 for oxymetazoline or noradrenaline, and slightly reduces the maximum efficacy of α_{1A} AR activation by oxymetazoline (**Supplementary Figs. 7a-d**), suggesting that Nb29 appears to antagonize receptor activation or possibly block the ligand entry into the orthosteric pocket. It should be noted that the radioligand competition assay was performed in equilibrium and the agonists had a longer incubation time to access the orthosteric pocket than in the signaling assay. In both assays, the effects of Nb29 are larger for oxymetazoline compared with noradrenaline, which is consistent with Nb29's binding selectivity towards the oxymetazoline-bound state of the α_{1A} AR in the titration assay (**Supplementary Figs. 1c-f and 7**). Thus, Nb29 might be considered a weak positive allosteric modulator or a neutral allosteric modulator, given that it binds to a known allosteric binding pocket in other GPCRs."

a. As it seems that all prazosin-like antagonists also bind in the exosite and ECL2 (Deluigi et al, 2022 Nature Commun.), NB29 may have effects on 3H-prazosin binding that are not allosteric. Verification of Increased affinity by ligand binding should be performed with 125I-HEAT.

We thank the reviewer for the suggestion. However, the previous mutagenesis such as I85T^{2.63}, F86M^{2.64}, E87A^{2.65} and F308A^{7.35} decreased ¹²⁵I-HEAT binding (Hamaguchi et al. 1998, Biochemistry, 5730-5737; Maiga et al. 2013 PLoS ONE, e68841). Thus, HEAT also seems to bind to the exosite.

We agree that the direct competition between Nb29 and 3H-prazosin may affect the agonist-competition curves. Since we cannot find a proper control that only competes with 3H-prazosin binding but not agonist binding, we cannot separate the potential allosteric effect from the competition effects. We acknowledged the limitation of these studies as noted above.

It should be mentioned that we observed a larger left-shift of the oxymetazoline competition curves compared to the noradrenaline competition curves, which agrees with the on-yeast titration results that Nb29 prefers the oxymetazoline-bound state of the α_{1A} AR but not the noradrenaline-bound state.

b. Was ligand binding performed at equilibrium? Was that even determined? Allosteric binding needs to be done at 4x equilibrium as artifacts appear and your binding differences are only 3- fold apart.

We incubated for 1 hour at room temperature for the competition ligand-binding assay. This incubation time is also used in previous studies (ref. 35) mentioned in major comment 2 above. In the revised manuscript, we found that ligand binding assay with purified α_{1A} AR exhibited more binding differences. Please see Supplementary Fig. 1e and f. It should be noted that we observe an approximately 34-fold increase in binding affinity for oxymetazoline when using a higher concentration of Nb29.

c. All the binding curves are normalized to their own binding max which may produce leftward shifts if Nb29 masks oxymetazoline binding or you are not at equilibrium. The binding curves should be normalized not to their own maximum but to the maximum binding (100%) of alpha1A-AR with oxymetazoline.

We performed the ligand binding assay with the purified α_{1A} AR-bound M1-Flag affinity resin for oxymetazoline (Oxy), noradrenaline (NA) or Nb29, and normalized to maximum binding of oxymetazoline binding. Please see Supplementary Fig. 1d-f.

d. A 3-fold shift in affinity is not very large to suggest a PAM. To confirm PAM effect would need to determine effects on signaling or other downstream effector.

We thank the reviewer for the suggestion. As mentioned above, we now indicate that "Nb29 might be considered a weak positive allosteric modulator or a neutral allosteric modulator, given that it binds to a known allosteric binding pocket in other GPCRs."

e. Is Nb29 + oxymetazoline binding effects specific to oxymetazoline or are similar effects seen with other ligands? Need to see effects with endogenous ligands also.

On-yeast titration assay indicated that Nb29 has a selectivity for the oxymetazoline-bound state compared with noradrenaline. This tendency is also observed in binding assay and signaling assay (Supplementary Fig. 1c-d and 7).

f. 3-fold differences call for more points on the curve. It appears that there are no error bars on the supplemental Fig 1d for alpha1A + Nb29. Is that a mistake or there really is no error in N=3 experiments?

We thank the reviewer for pointing out this. There are error bars on supplemental Fig 1d. In the revised figure, we modified the symbol's shape and size (from square to circle) and the thickness of the line to make the error bars clearer.

Minor Comments:

1. Why was oxymetazoline used for the structure as opposed to the better selective imidazoline agonists, such as cirazoline or A61603? Oxymetazoline is a very poor agonist at the α_1A -AR. Was other agonists attempted but oxymetazoline gave the better crystal?

As there have been no structure of α_{1A} AR, we initially would like to solve the structure bound to the clinically used drug. Oxymetazoline is used for the treatment of nasal congestion and is much cheaper than A61603. For receptor expression and purification, we added the ligand to stabilize the receptor.

2. In fig 4e, it appears that TM3 of the α_1B -AR structure has a more inward tilt than in the α_1A -AR. Please comment. Is it consequential?

We added the comment and revised Fig4c and its legend. Note that, inactive α_{1B} AR structure (Deluigi et al, 2022, Nat Comm; PDB ID: 7B6W) was determined by thermostabilizing α_{1B} AR mutant, which contains S95C^{2,54}, S150Y^{34,50}, G183V^{4,63}, D191Y^{ECL2}, T295M^{6,36}, V333L^{7,38}, F334L^{7,39} and P349L^{7,54}. Among them, F334L^{7,39} seems to allow the (+)cyclazosin to shift towards TM7, leading to inward movement of TM3 of α_{1B} AR. Please see in line 274.

3. Line 48- I am unaware of any α_1A/D designation being generally accepted or published at the time. There was an α_1A/C designation that Paul Simpson published several manuscripts with that designation. The past nomenclature is confusing enough for the non-adrenergic scientists, maybe best to leave that out (α_1A/D)?

We thank the reviewer for this helpful suggestion. We suppose that the description of why there is no α_{1C} AR might be useful information for some general readers who are not experts on adrenergic receptors. We have revised this to the “formerly α_{1A} AR and α_{1D} AR” in line 48.

4. S188 was identified as responsible for α_1A -AR agonism first in this publication, “Hwa J, Perez DM. The unique nature of the serine interactions for alpha 1-adrenergic receptor agonist binding and activation. J Biol Chem. 1996 Mar 15;271(11):6322-7”, which you do not reference. Although this publication suggested that the meta-hydroxyl interacted with S188 because phenylephrine, a full agonist, only contains a meta-hydroxyl, the main point of this publication was that S188 was responsible for agonism which is confirmed in your structure.

We thank the reviewer for this helpful suggestion. This publication was added to reference 34. Please see line 181.

5. Line 227-8. The “Hwa J, Perez DM. The unique nature of the serine interactions for alpha 1-adrenergic receptor agonist binding and activation. J Biol Chem. 1996 Mar 15;271(11):6322-7”, publication also first mentions the differences in the catechol ring orientation (about 120 degrees) between α_1 and β_1 -ARs. I couldn't tell the degree difference in your figures for the catechol ring, but please ref this publication.

We thank the reviewer for this helpful suggestion. The publication was added as reference 34 in line 232. As for the catechol ring orientation, a structural comparison between α_{1A} AR (green) bound noradrenaline (grey) and β_1 AR (pink) bound noradrenaline (yellow) related to Fig 4b is shown below. (Left: side view, right: top view).

6. Imidazolines have been identified to bias signaling towards Gs for the α_1A -AR (Evans et al.,

molpharm 79: 298, 2011; da Silva et al., Br. J. Pharmacol, 174: 2318, 2017). Is there any indication for this in your model?

We thank the reviewer for raising the interesting point of the Gs bias signaling. Unfortunately, our structures, however, do not provide any insights into this signaling bias.

Reviewer #2 (Remarks to the Author):

Toyoda et al describe three new cryo-EM structures of the alpha1a-adrenergic receptor (alpha1AR), two coupled to a G protein in an active conformation and one in the inactive state. The two active state structures are bound either to the native agonist noradrenaline or to the selective agonist oxymetazoline. The inactive state structure was possible only through a combination of two stabilising mutations, engineering the receptor to bind Nb6 on the intracellular surface and binding of a newly selected nanobody to the extracellular surface (Nb29). The structures are of sufficient quality to substantiate the claims made by the authors in the manuscript on ligand binding and the activation mechanism.

The manuscript describes in great detail each of the three structures, how the ligands bind and how they differ from other alpha and beta ARs. The activation mechanism of the receptors is directly analogous to other Class A receptors. The structures provide a basis for understanding ligand selectivity between the different receptors. In addition, Nb29 is an interesting addition to the range of nanobodies identified to date that bind to the extracellular surface of GPCRs as it stabilizes the conformation of the inactive state and therefore behaves like a PAM.

We thank the reviewer for the positive comments on our work and appreciate the suggestions that helped us improve the manuscript.

The manuscript is well written and presented and I have only a few minor comments that need addressing.

1. Line 96, Supplementary Figs 2b and 2c are not mentioned in the main text.

We added Supplementary Figs 2b and 2c in line 105.

2. Line 118, insert ‘...had been determined’ after ‘complex’?

We revised from “As the cryo-EM...” to “Based on the cryo-EM...”. Please see line 125.

3. Line 144, states between which atoms the distance of 14.5Å was measured.

We measured the distance α -carbon of E269^{6.30} (E269L^{6.30} in α_{1A} AR-kOR) in TM6, as E268^{6.30} was measured in β_2 AR-Gs complex (Rasmussen et al., 2011, Nature, 549-555). We added the description of the distances in the legend of Fig. 2d.

4. Line 224, It is mentioned that the para-hydroxyl of the catechol hydrogen bonds to Ser5.42 in both alpha1AR and alpha2AR, suggesting the mode of catechol binding is similar. However, because the rotamer of Ser5.42 differs between the two receptors, then there is a distinct difference in the pose of the catechols. Perhaps this would be good to mention.

We mentioned this in line 239.

5. Line 227, delete ‘more’

We deleted “more” from the previous sentence “The noradrenaline binding pose of β_1 AR is ~~more~~ different from α_{1A} AR” in line 236.

6. Line 327, define ‘ligand’ as agonist or antagonist

Alternatively, we added “leading to enhance antagonist binding” after mentioning EP4-Fab. By adding this, all three receptors, AT2, EP4 and D2R, were defined as the agonist or antagonist binding. Please see line 367.

7. Line 328, the words ‘diagonally’ and ‘upright’ are meaningless without a context. Reformulate.

We deleted these words. Please see line 365.

8. Lines 357 and 387, you cannot say ‘may be involved in the G protein coupling selectivity’ as you have

provided insufficient data directly pertinent to the role of the specific residues in this context e.g. mutagenesis data of specific residues that alter the coupling specificity. Please delete.

We deleted the last half of this sentence; “These residues of α_{1A} AR are not conserved in α_2 ARs and β ARs, and therefore may be involved in the G-protein coupling selectivity.” Please see line 397.

9. Line 700, define unmodelled grey density in Figs 1a, b and c.

We added, “The detergent micelle (a, c and e) and unmodelled mT4L (b) are shown in grey.”, in line 774.

10. Fig 4b. The hydrogen bonds between noradrenaline and the side chains of alpha 1AR are missing. Please add.

We added the bonds between α_{1A} AR and noradrenaline in Fig.4b.

11. Supplementary Figs 1, 2 and 4. Please define in the legends the number of technical and biological replicates for the binding data (not just ‘n=3 measurements’ which is ambiguous). Also, give the errors for all the K_i values and define what they are.

We added the error values in the legends of Supplementary Figs 1, 2 and 4.

12. Supplementary Fig 3, and Fig 5g,h, and Fig 1b,d,f. Give sigma values and cut-offs for the density mesh, and which program was used.

For Supplementary Fig 3, 4h and 5g,h, the density map are contoured at 1.5σ , using the isomesh command in pymol. For Fig.1d-f, the density maps are manually adjusted by UCSF ChimeraX.

13. Supplementary Fig 6. There is no panel ‘b’, so it seems superfluous to label this ‘a’. What do the red amino acid residues mean compared to the black (also in Sup Fig 9)?

We deleted the label “a” from Supplementary Fig6. Conserved amino acids with α_1 AR at the same position were shown as a red character. We added this comment in the legend of Supplementary Fig. 6.

14. Supplementary Fig 7. What do the triangles mean in panel b?

Amino acids involved in hydrogen bonds and salt bridges between α_1 AR and Nb29 are marked with black and orange triangles, respectively. We added this sentence in the figure legend of Supplementary Fig. 8b.

Reviewer #3 (Remarks to the Author):

The manuscript by Toyoda et al presents the cryoEM structures of the human GPCR alpha-1a adrenergic receptor that responds to adrenaline and noradrenaline and is involved in smooth muscle contraction and cognitive function. Three conformations were solved. First, the active conformation in complex with the miniGq protein bound either to the endogenous agonist (noradrenaline) or a selective agonist of the alpha-1a receptor (oxymetazoline) and stabilized by the nanobody (VHH) Nb29 that binds to the extracellular part of the receptor. The third structure reports the inactive conformation of the alpha-1a adrenergic receptor in complex with another nanobody (Nb6) that binds to the intracellular part of the receptor.

The results are original since in the alpha-1 receptors family only the crystal structure of the alpha-1b receptor was previously reported (Deluigi et al (2022) Nat Comm). The selectivity of ligand binding was largely reported in the study.

In addition, in the present work the author discovered the Nb29 nanobody as a positive allosteric modulator that specifically binds to the extracellular vestibule of the alpha-1a receptor.

We thank the reviewer for the positive comments on our work and appreciate the suggestions that helped us improve our manuscript.

Comments:

1. Do the nanobodies Nb29 and Nb6 are required to stabilize the active and inactive conformations, respectively, of the alpha-1a receptor?

Nb29 preferentially binds the oxymetazoline-bound receptor. The miniGsq protein is the key to stabilizing the active conformation. In addition, an important role of Nb29 is to help with particle alignment during cryo-EM data processing.

Nb6 also helps the receptor to stabilize in an inactive conformation, however, the antagonist-bound receptor already prefers inactive conformation. The most important role of Nb6 is to work as a fiducial mark and help particle alignment during cryo-EM data processing.

2. Evidence for positive allosteric modulation of the Nb29 is shown only by binding experiments. It would be even more compelling if functional data using signaling assays were provided. Does the Nb29 alone (absence of alpha-1 receptor agonist) could active the receptor?

As noted above, in the revised manuscript we include the following paragraph to qualify our definition of Nb29 as an allosteric modulator.

“Nb29 binds to the extracellular side of α_{1A} AR which is topologically distinct from the orthosteric agonist pocket (**Fig. 1**). This site has been shown to bind to allosteric modulators for muscarinic receptors (ref). We do observe a left shift of the agonist competition binding curves in the presence of Nb29 (**Supplementary Figs. 1d-f**); however, these experiments are complicated by the fact that Nb29 is a competitive inhibitor of the radioligand [3 H] prazosin. In cell signaling assays, Nb29 exhibits no agonist activity on its own, has no effect on EC50 for oxymetazoline or noradrenaline, and slightly reduces the maximum efficacy of α_{1A} AR activation by oxymetazoline (**Supplementary Figs. 7a-d**), suggesting that Nb29 appears to antagonize receptor activation or possibly block the ligand entry into the orthosteric pocket. It should be noted that the radioligand competition assay was performed in equilibrium and the agonists had a longer incubation time to access the orthosteric pocket than in the signaling assay. In both assays, the effects of Nb29 are larger for oxymetazoline compared with noradrenaline, which is consistent with Nb29’s binding selectivity towards the oxymetazoline-bound state of the α_{1A} AR in the titration assay (**Supplementary Figs. 1c-f and 7**). Thus, Nb29 might be considered a weak positive allosteric modulator or a neutral allosteric modulator, given that it binds to a known allosteric binding pocket in other GPCRs.”

3. What is the affinity of the nanobody Nb29?

We referred to the previous report of the synthetic nanobody library (Supplementary Fig. 5 of McMahon et al. Nat Struct Mol Biol, 2018) and performed the on-yeast titration assay by flow cytometry in Supplementary Fig. 1c. Nb29 selectively binds the oxymetazoline-bound α_{1A} AR over apo and antagonist-bound α_{1A} AR. K_d values are 57.4 ± 16.5 nM for the oxymetazoline-bound state.

4. Which alpha-1 receptor constructs are using for nanobody screening (line 463)? It should be clarified.

We used the C-terminus truncated human α_{1A} AR (residues 1–370, full length: 466). We mentioned this in the method section of the nanobody screening, line 502.

5. Line 76: “class A” should be added since the structure of a nanobody in complex with the metabotropic glutamate mGlu2 receptor was reported by the Beili Wu’s group (Lin et al (2021) Nature).

We added, “class A” in line 75.

6. Line 296: repetition of line 283.

We deleted line 296 in the previous text and revised the Nb29 paragraph. Please see line 314 in the revised manuscript as mentioned in our answer 2 above.

REVIEWER COMMENTS

Reviewer #1 (Remarks to the Author):

The authors have responded to my previous comments concerns. The manuscript can be accepted for publication.

Reviewer #2 (Remarks to the Author):

The authors have addressed all comments satisfactorily and I have no further suggestions for the manuscript

Reviewer #3 (Remarks to the Author):

The manuscript by Toyoda et al reports three cryo-EM structures of the alpha-1a adrenergic receptor (Alpha1aAR): two active states bound to the endogenous noradrenaline, or the partial agonist oxymetazoline that is used clinically for the treatment of nasal congestion, in complex with the G protein construct miniGsq, and the nanobody Nb29 bound to the extracellular part of the receptor; an inactive state bound to the antagonist tamsulosin and stabilized by the nanobody Nb6 that binds to the intracellular side of the receptor.

This study is original because alpha-1 adrenergic receptor structures are important due to the lack of selective ligands for these Alpha1 receptor subtypes, and so far only one crystal structure has reported the inactive state of the Alpha1aAR in the complex with an inverse agonist.

The study is mainly descriptive, except for experiments to investigate the effect of the nanobody Nb29 on the binding of the antagonist and on the coupling with the G protein. No mutations were performed. But the manuscript is well written.

Several important issues should be clarified. Especially, the effect of the nanobody Nb29 on the receptor is not clear.

Major points:

- Line 91-92: "high selectivity for oxymetazoline-bound state" is overstated. Indeed, the affinity of the nanobody Nb29 is a factor two better for oxymetazoline than for other ligands.
- Line 93-94: the Nb29-induced left shift induced is large (more than one Log) with the purified receptor, but much smaller (less than one Log) with the receptor in Sf9 membranes. Why?
- Line 301-304: the larger effect of the nanobody Nb29 on oxymetazoline compared to noradrenaline might be better explained by the fact that oxymetazoline is a partial agonist, than by the two-fold difference in affinity between these two agonists.
- Line 304-305: it is not clear whether Nb29 is a PAM or a neutral allosteric modulator. The signaling assay used is the Glo sensor (kinetic assay). The effects of the nanobody on signaling would have been easier to characterize using an end-point assay (cAMP accumulation).
- Signaling assay: Alpha1aAR is primarily coupled to Gq, why not use an intracellular calcium readout or the IP1 accumulation assay (HTRF) to characterize the effect of the Nb29 nanobody.
- G protein constructs used in this study: why use the Gq/s chimera for the mini-G and not mini-Gq? This should be clarified in the text.
- Line 157-158: the side chain displacement of W285 6.48 is small. Can you give a distance value?

Minor points:

- Supp Figure 7d: the Y-axis legend should be clarified to better understand what is measured.
- Line 93: "purified" should be removed.
- Line 119-120: please, clarify this sentence. What is "the N29-dissociated complex"?
- Line 349-350: this sentence should be clarified. Why "more frequently targeted"?
- Typographical errors: "c-terminus" (line 522); "NaCl" (line 539); remove "," (line 550).

Point-by-point response to the reviewers' comments

Reviewer #1 (Remarks to the Author):

The authors have responded to my previous comments concerns. The manuscript can be accepted for publication.

We thank the reviewer for accepting our manuscript.

Reviewer #2 (Remarks to the Author):

The authors have addressed all comments satisfactorily and I have no further suggestions for the manuscript

We thank the reviewer for accepting our manuscript.

Reviewer #3 (Remarks to the Author):

The manuscript by Toyoda et al reports three cryo-EM structures of the alpha-1a adrenergic receptor (Alpha1aAR): two active states bound to the endogenous noradrenaline, or the partial agonist oxymetazoline that is used clinically for the treatment of nasal congestion, in complex with the G protein construct miniGsq, and the nanobody Nb29 bound to the extracellular part of the receptor; an inactive state bound to the antagonist tamsulosin and stabilized by the nanobody Nb6 that binds to the intracellular side of the receptor.

This study is original because alpha-1 adrenergic receptor structures are important due to the lack of selective ligands for these Alpha1 receptor subtypes, and so far only one crystal structure has reported the inactive state of the Alpha1aAR in the complex with an inverse agonist.

The study is mainly descriptive, except for experiments to investigate the effect of the nanobody Nb29 on the binding of the antagonist and on the coupling with the G protein. No mutations were performed. But the manuscript is well written.

Several important issues should be clarified. Especially, the effect of the nanobody Nb29 on the receptor is not clear.

We thank the reviewer for the comments on our revised work.

Major points:

- Line 91-92: "high selectivity for oxymetazoline-bound state" is overstated. Indeed, the affinity of the nanobody Nb29 is a factor two better for oxymetazoline than for other ligands.

We deleted "high" from this sentence. Please see line 91.

- Line 93-94: the Nb29-induced left shift induced is large (more than one Log) with the purified receptor, but much smaller (less than one Log) with the receptor in Sf9 membranes. Why?

We don't know the clear reason but suppose that this difference is derived from the surrounding environment of the receptor. Nb29 was screened using the purified α_{1A} AR in detergent micelles of MNG and CHS (line 505 in Method), suggesting that Nb29 is more effective against the purified receptor (Supp Fig 1b-c and 1e-f). The cell membranes contain several phospholipids, fatty acids and other endogenous proteins, which might reduce the agonist binding of α_{1A} AR in the cell membrane (Supp Fig 1d).

- Line 301-304: the larger effect of the nanobody Nb29 on oxymetazoline compared to noradrenaline might be better explained by the fact that oxymetazoline is a partial agonist, than by the two-fold difference in affinity between these two agonists.

We thank the comment. Although partial agonism activity by S188^{5,42} (line 190) might be involved in the Nb29's effect, it is too far from the Nb29 binding site compared with F305^{7,35}. In addition, future structural and functional studies using various types of ligands might provide further information.

- Line 304-305: it is not clear whether Nb29 is a PAM or a neutral allosteric modulator. The signaling assay used is the Glo sensor (kinetic assay). The effects of the nanobody on signaling would have been easier to characterize using an end-point assay (cAMP accumulation).

As described in the "Structural insight into Nb29 binding" section, we think that it's hard to define whether Nb29 is a PAM or neutral allosteric modulator because Nb29's function is complicated. Thus, we revised the paper title from "allosteric modulation" in the initial submission to "nanobody recognition" in the previous-revised manuscript.

Supplementary Figs 7a-c display the end-point of cAMP accumulation at 10 min after stimulation. The downward curve shift was also observed after 30 min.

- Signaling assay: Alpha1aAR is primarily coupled to Gq, why not use an intracellular calcium readout or the IP1 accumulation assay (HTRF) to characterize the effect of the Nb29 nanobody.

Our initial attempts of Gq-dependent signaling assays such as Ca²⁺ Fluo-4 direct assay, IP-One assay and Nanobit G protein dissociation assay were not worked well due to poor signals, thus we performed Glo-sensor cAMP with Gsq protein, instead.

- G protein constructs used in this study: why use the Gq/s chimera for the mini-G and not mini-Gq? This should be clarified in the text.

Reference 29 (Nehme et al. *PLoS One*, 2017) reported that mini-Gq was not suitable for expression in *E. coli*, thus they designed miniGsq. We added this explanation in line 534.

- Line 157-158: the side chain displacement of W285 6.48 is small. Can you give a distance value?

The maximum distances of the side chain displacement of W285 (position 7th carbon of the indole ring) are 2.4 Å between the noradrenaline-bound active state and the tamsulosin-bound inactive state, and 1.8 Å between the oxymetazoline-bound state and the tamsulosin-bound state. We added this sentence in the Figure 2 legend (line 791).

Minor points:

- Supp Figure 7d: the Y-axis legend should be clarified to better understand what is measured.

We revised from “Noradrenaline %” to “ α_{1A} AR activation (Noradrenaline %)” in Supp Figure 7d.

- Line 93: “purified” should be removed.

We removed it from the text. Please see line 93.

- Line 119-120: please, clarify this sentence. What is “the N29-dissociated complex”?

We revised it to “the Nb29-dissociated α_{1A} AR-miniGsq complex” (line 119).

- Line 349-350: this sentence should be clarified. Why “more frequently targeted”?

We revised this sentence from “peptide-binding GPCRs are more frequently targeted by antibodies because of their large binding pockets.” to “peptide-binding GPCRs are more frequently targeted by antibodies because they have relatively large binding pockets compared with the small-molecule binding GPCRs such as aminergic GPCRs.” Please see line 350.

- Typographical errors: “c-terminus” (line 522); “NaCl” (line 539); remove “,” (line 550).

We revised “C-terminus (line 524)”; NaCl (lines 535 and 542); removed “,”